# Cre-assisted fine-mapping of neural circuits using orthogonal split inteins

**Haojiang Luan[1], Alexander Kuzin[2], Ward F Odenwald[2], Benjamin H White[1]\***

[1]Laboratory of Molecular Biology, National Institute of Mental Health, NIH, Bethesda, United States; [2]Neural Cell-Fate Determinants Section, National Institute of Neurological Disorders and Stroke, NIH, Bethesda, United States

**Abstract** Existing genetic methods of neuronal targeting do not routinely achieve the resolution required for mapping brain circuits. New approaches are thus necessary. Here, we introduce a method for refined neuronal targeting that can be applied iteratively. Restriction achieved at the first step can be further refined in a second step, if necessary. The method relies on first isolating neurons within a targeted group (i.e. Gal4 pattern) according to their developmental lineages, and then intersectionally limiting the number of lineages by selecting only those in which two distinct neuroblast enhancers are active. The neuroblast enhancers drive expression of split Cre recombinase fragments. These are fused to non-interacting pairs of split inteins, which ensure reconstitution of active Cre when all fragments are expressed in the same neuroblast. Active Cre renders all neuroblast-derived cells in a lineage permissive for Gal4 activity. We demonstrate how this system can facilitate neural circuit-mapping in *Drosophila*.

## Introduction

An essential step in mapping brain circuits is identifying the function of the individual neurons that comprise them. This is commonly achieved by manipulating neuronal function using effectors encoded by transgenes whose expression is targeted to small subsets of cells using the regulatory elements of neurally-expressed genes (*Gohl et al., 2017*; *Luo et al., 2018*). While it has proved relatively easy to target large groups of neurons for cellular manipulation by this means in genetic model organisms using binary expression systems, such as the Cre-lox system of mice or the Gal4-UAS system of fruit flies, highly-specific targeting of neurons requires combinatorial methods. Genetic combinatorial methods typically use either the regulatory elements of two neurally-expressed genes or exploit stochastic events to limit transgene targeting to a subpopulation of a larger group of neurons. In fruit flies, both types of method have been used to target single cells under optimal conditions (*Aso et al., 2014*; *Gao et al., 2008*; *Gordon and Scott, 2009*; *Kohatsu et al., 2011*; *Lee and Luo, 1999*; *Luan et al., 2012*; *Pool et al., 2014*; *Sen et al., 2019*; *Shang et al., 2008*; *Yu et al., 2010*), and combinatorial approaches using three regulatory elements have achieved some success (*Dolan et al., 2017*; *Pankova and Borst, 2017*; *Shirangi et al., 2016*; *Sullivan et al., 2019*). However, general limitations apply to all such approaches: stochastic methods are, by nature, poorly reproducible, while combinatorial methods are labor-intensive, often requiring the characterization of many neurally active enhancer elements (*Dionne et al., 2018*; *Tirian et al., 2017*). Simpler methods of targeting small populations of brain cells are therefore desirable in the effort to comprehensively map neural function.

An attractive approach to increase the specificity of neuronal targeting is to identify neurons based not only on the genes they express in the terminally differentiated state (i.e. terminal effector genes, TEG), but also on their developmental history (*Awasaki et al., 2014*; *Dymecki et al., 2010*; *Huang, 2014*). Most neuronal lineages produce diverse neuron types, and while some striking correspondences have been found (*Lacin et al., 2019*), lineage identity, in general, correlates poorly with

**\*For correspondence:**
benjaminwhite@mail.nih.gov

**Competing interests:** The authors declare that no competing interests exist.

**eLife digest** In humans – as well as flies and most other animals – the brain controls how we move and behave, and regulates heartbeat, breathing and other core processes. To perform these different roles, cells known as neurons form large networks that quickly carry messages around the brain and to other parts of the body. In order to fully understand how the brain works, it is important to first understand how individual neurons connect to each other and operate within these networks.

Fruit flies and other animals with small brains are often used as models to study how the brain works. There are several methods currently available that allow researchers to manipulate small groups of fruit fly neurons for study, and in some cases it is even possible to target individual neurons. However, it remains an aspirational goal to be able to target every neuron in the fly brain individually.

The Gal4-UAS system is a way of manipulating gene activity widely used to study neurons in fruit flies. The system consists of two parts: a protein that can bind DNA and control the activity of genes (Gal4); and a genetic sequence (the UAS) that tells Gal4 where to bind and therefore which genes to activate. Fruit flies can be genetically engineered so that only specific cells make Gal4. This makes it possible, for example, to limit the activity of a gene under the control of the UAS to a specific set of neurons and therefore to identify or target these neurons. Luan et al. developed a new technique named SpaRCLIn that allows the targeting of a subset of neurons within a group already identified with the Gal4-UAS system.

During embryonic development, all neurons originate from a small pool of cells called neuroblasts, and it is possible to target the descendants of particular neuroblasts. SpaRCLIn exploits this strategy to limit the activity of Gal4 to smaller and smaller numbers of neuroblast descendants. In this way, Luan et al. found that SpaRCLIn was routinely capable of limiting patterns of Gal4 activity to one, or a few, neurons at a time. Further experiments used SpaRCLIn to identify two pairs of neurons that trigger a well-known feeding behavior in fruit flies. Luan et al. also developed a SpaRCLIn toolkit that will form the basis of a community resource other researchers can use to study neurons in fruit flies. These findings could also benefit researchers developing similar tools in mice and other animals.

neuronal identity as defined by gene expression (*Hobert et al., 2016*; *Zeng and Sanes, 2017*). Conversely, gene expression is often correlated across neurons that differ in identity as defined by their function, morphology, and neuroanatomical location (*Hobert, 2016*; *Hobert and Kratsios, 2019*). This is because neuronal identities are defined not by single genes, but by the expression of often overlapping batteries of TEGs. An intersection of lineage with the expression of a specific TEG may thus, in general, include fewer neurons than an intersection of the expression patterns of two TEGs. In addition, because neurons from a given lineage typically remain regionally localized, intersections made using lineage information will tend to restrict neuronal targeting anatomically.

Recombinase-based intersectional methods that combine information about lineage and cell type have been developed in both mice and fruit flies and have been shown to substantially restrict targeting to cell groups of interest (*Brust et al., 2014*; *Ren et al., 2016*). However, the use of such methods has remained largely limited to specific cases—in mice, sublineages of brainstem serotonergic neurons (*Okaty et al., 2015*), and in flies, subtypes of Type II transit-amplifying neural stem cells (i.e. neuroblasts, NBs) of the central brain (*Ren et al., 2018*; *Ren et al., 2017*). This is because of the paucity of lineage-restricted enhancers. Just as there are few TEG enhancers that are active in small numbers of mature neurons, there are also few identified enhancers that exhibit lineage-specific activity. In the fly, a systematic analysis of some 5000 neural enhancer domains identified 761 with activity in embryonic NBs, but 99 of these expressed in most or all lineages (*Manning et al., 2012*). A separate analysis indicates that the remainder are at best active in 5–20 lineages (*Awasaki et al., 2014*). The routine use of lineage-cell type intersections for neural circuit mapping will thus require more refined methods of isolating neuronal lineages or sub-lineages.

To achieve such lineage refinement, we introduce here a combinatorial method analogous to the Split Gal4 technique used to restrict neuronal targeting to the intersection of two TEG expression

patterns (*Luan et al., 2006*). We restrict reconstitution of a Split Cre recombinase to the expression patterns of two independent NB-active enhancers (i.e. NBEs). Only NBs in which both enhancers are active thus make full-length Cre. Cre is then used to selectively promote activity of the Gal4 transcription factor—expressed under the control of a TEG enhancer—in the mature progeny of these NBs, thus implementing a second intersection. Our method (i.e. 'Split Cre-assisted Restriction of Cell Class-Lineage Intersections,' or SpaRCLIn) generalizes the capabilities of the CLIn technique introduced by *Ren et al. (2016)* by expanding the range of possible intersections to most *Drosophila* lineages while maintaining compatibility with all existing *Drosophila* Gal4 driver lines. To facilitate SpaRCLIn's use, we have generated a variety of tools, including two libraries of transgenic fly lines, each of which expresses distinct Split Cre components under the control of 134 different NBEs. We characterize the efficacy of these SpaRCLIn reagents and provide examples of their use in restricted neuronal targeting and circuit-mapping.

## Results

### Development of bipartite and tripartite split cre recombinases

SpaRCLIn was developed to refine the expression pattern of a Gal4 driver using the basic strategy shown in *Figure 1*. In common with other existing methodologies, SpaRCLIn uses a recombinase (i.e. Cre) to excise an otherwise ubiquitously expressed construct encoding Gal80, a suppressor of the Gal4 transcription factor (*Figure 1A–B*). As in the CLIn technique, recombinase expression—and thus the excision of Gal80—occurs only in targeted NBs, rendering the progeny of these NBs permissive to Gal4 activity (*Figure 1C*). Those progeny that lie within the expression pattern of the Gal4 driver will be competent to drive UAS-reporters and effectors, such as UAS-GFP. In the SpaRCLIn technique, distinct NBEs are used to express components of a bipartite Split Cre molecule in restricted subsets of NBs. In lineages of these NBs that contain mature neurons within the Gal4 expression pattern, Gal4 will be active. This population of neurons can be additionally parsed using a tripartite Split Cre to further restrict the subset of NBs that make active Cre (*Figure 1D*).

Although most recombinase-based expression systems in *Drosophila*, such as MARCM (*Lee and Luo, 1999*), Flp-out Gal80 (*Gordon and Scott, 2009*), and FINGR (*Bohm et al., 2010*) have preferentially used the Flp recombinase for Gal80 excision, we selected Cre for use in SpaRCLIn because of its demonstrated ability to retain high activity in a variety of bipartite forms (*Hirrlinger et al., 2009*; *Jullien, 2003*; *Kawano et al., 2016*; *Kennedy et al., 2010*; *Rajaee and Ow, 2017*). Although Cre activity has been reported to be toxic in *Drosophila* when chronically expressed at high levels (*Heidmann and Lehner, 2001*; *Nern et al., 2011*), it has previously been used in NBs without apparent adverse effects (*Awasaki et al., 2014*; *Hampel et al., 2011*; *Ren et al., 2016*). Because our system requires use of a tripartite Cre to achieve the most refined targeting it was also desirable to use a method of splitting Cre that would permit reconstitution of the intact molecule to obtain the highest activity levels. Split inteins, which are capable of autocatalytically joining two proteins to which they are fused, are well-suited to this purpose and distinct split inteins have been previously shown to support reconstitution of recombinase activity from complementary Cre fragments fused to them (*Ge et al., 2016*; *Han et al., 2013*; *Hermann et al., 2014*; *Wang et al., 2012*). *Figure 1E* shows the primary structure of Cre, indicating the location of the breakpoints (green highlight) at which we introduced split intein moieties into the molecule. These breakpoints separate the amino acid residues in the primary structure that form the DNA-binding sites (blue) and the active site (yellow highlight), thus insuring that none of the fragments retains catalytic activity. Two Split Cre fragments, $Cre_{AB}$ and $Cre_C$, were generated by the breakpoint between amino acids P250 and S251 to implement the bipartite Split Cre system (*Figure 1F,G*), while dividing the $Cre_{AB}$ fragment at the breakpoint between amino acids D109 and S110 was used to create two further fragments (i.e. $Cre_A$ and $Cre_B$) which together with $Cre_C$ form the basis of the tripartite Split Cre system (*Figure 1H,I*). The split intein pairs used to generate these fragments, gp41-1 and NrdJ-1, were chosen based on their trans-splicing efficiency and their lack of cross-reactivity (*Carvajal-Vallejos et al., 2012*). The latter criterion was critical for avoiding the generation of unproductive fusion products of the Cre fragments.

After confirming the ability of the bi- and tripartite constructs to reconstitute Cre activity when co-expressed in transfected S2 cells (data not shown), we used them to generate transgenic fly lines

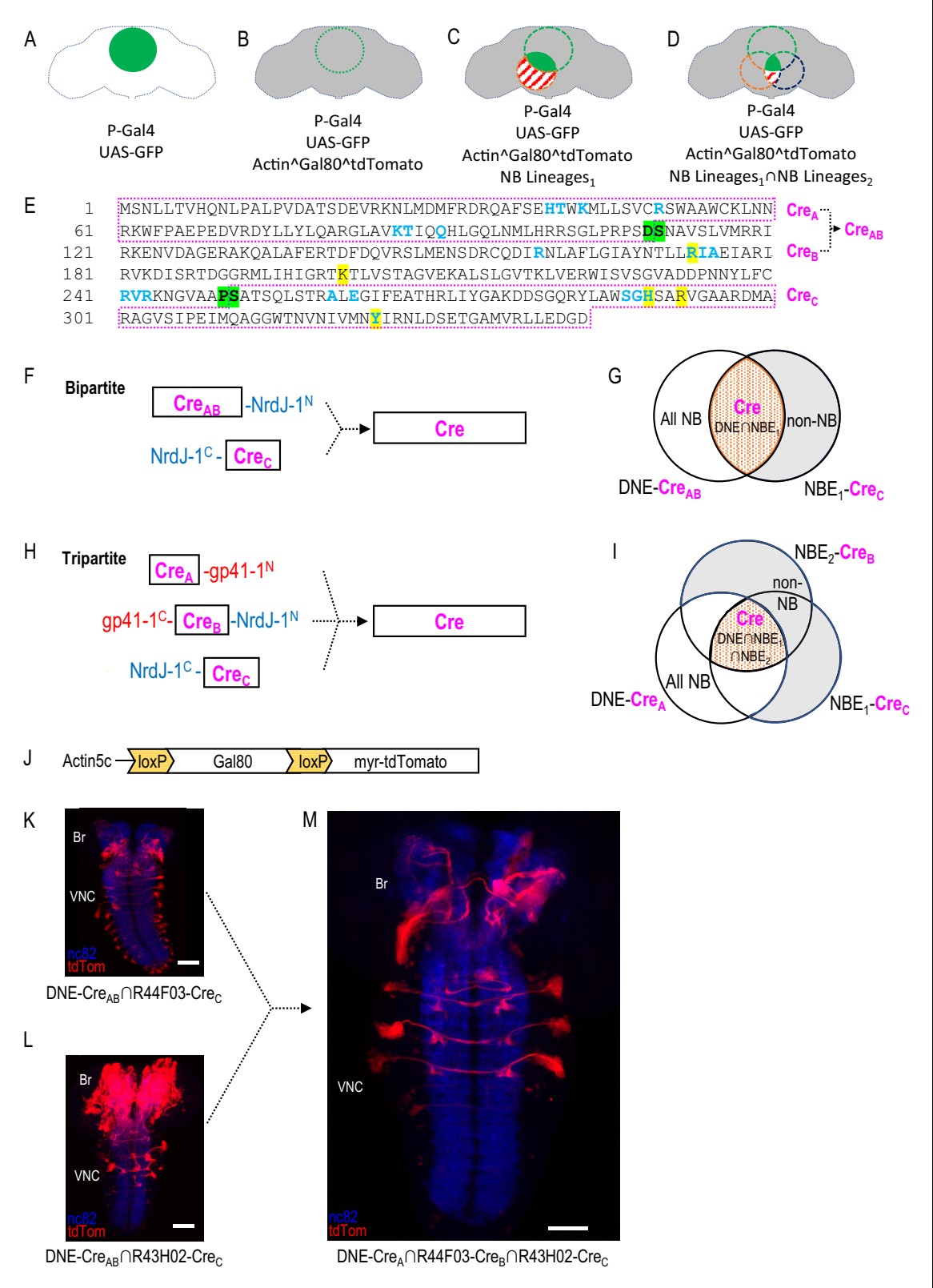

**Figure 1.** Restriction of NB targeting using split Cre components fused to split inteins. (A–D) Components and genetic logic of the SpaRCLIn system. (A) A Gal4 driver that drives expression of UAS-transgenes, such as UAS-GFP, in a specific pattern of cells within the CNS (green filled circle). (B) Conditional expression of Gal80, a repressor of Gal4 activity, in all cells using an Actin5C promoter, subject to excision by Cre (gray shading indicates repression of Gal4 by Gal80). (C) Selective activation of Cre in specific NBs (red dotted circle) to excise Gal80 and permit expression of the marker

*Figure 1 continued on next page*

*Figure 1 continued*

tdTomato (red stripes) and activity of Gal4 (solid green) in neurons derived from those NBs. (D) Use of split Cre components to target NBs at the intersection of two NB expression patterns (red and blued dotted circles) to permit Gal4 activity selectively within cells derived from these NBs (solid green). Note that Gal80 excision results in persistent expression of tdTomato in the affected neurons, but that expression of UAS-reporters and effectors is determined by the temporal properties of the Gal4 line used to drive it. (E) Primary sequence of the Cre protein using the single letter amino acid code. Residues that participate in DNA-binding (blue) or catalysis (yellow highlight) are indicated as are the break-points (green highlight) chosen to generate the split Cre fragments for fusion to split inteins: $Cre_A$, $Cre_B$, $Cre_{AB}$, and $Cre_C$ as indicated (magenta boxes). (F–G) The bipartite SpaRCLIn system. (F) Schematics of the Cre fragments fused to NrdJ-1 split inteins, indicating their ability to reconstitute full-length Cre, (G) $Cre_{AB}$ expression is directed to all NBs (white plus red shading) using the NB-specific DNE enhancer (see text), and $Cre_C$ expression is directed to a subset of NBs (red) by the NBE enhancer, which will also express in other cell types (gray). Only the NBs targeted by NBE will express both $Cre_{AB}$ and $Cre_C$ and reconstitute full-length Cre. (H–I) The tripartite SpaRCLIn system. (H) Similar to (F) except that the CreAB fragment has been further divided into $Cre_A$ and $Cre_B$ components which have been fused to gp41-1 split inteins at breakpoints. All three fragments are now required to reconstitute full-length Cre. (I) Venn diagram similar to (G) indicating the intersection of the three enhancers used to drive $Cre_A$ (DNE), $Cre_B$ ($NBE_2$), and $Cre_C$ ($NBE_1$). (J) Schematic of the floxed Gal80 construct used in the SpaRCLIn system, the expression of which is driven by the ubiquitously active Actin5C promoter. Cre-mediated excision of Gal80 via the flanking loxP sites causes a myristoylated tdTomato (tdTom) red fluorescent protein to be expressed instead of Gal80. (K–M) Restriction of NB expression by SpaRCLIn. (K, L) tdTom expression (red) driven by the bipartite SpaRCLIn system using two different NBEs (44F03 and 43H02) to drive CreC expression. (M) tdTom expression driven by the tripartite SpaRCLIn system at the intersection of the two NBE expression patterns, which overlap in several NB pairs of the ventral nerve cord (VNC) and brain (Br). Neuropil labeling by the nc82 antibody is shown in blue. Scale bar: 50 μM. Note that the genotypes of the flies for panels of this and all subsequent figures are provided in *Supplementary file 2*. The online version of this article includes the following figure supplement(s) for figure 1:

**Figure supplement 1.** Expression patterns of neuroblast-active enhancers.

**Figure supplement 2.** A neuroblast-specific *deadpan-nerfin-1* enhancer, DNE.

**Figure supplement 3.** Reproducibility of expression patterns of NBE-CreB and -CreC lines.

in which they were expressed in patterns dictated by individual enhancers that exhibited activity in neuroblasts. Most of the NBEs selected for this purpose were taken from the large collection of enhancer fragments with fully defined sequences created by the Rubin lab (*Pfeiffer et al., 2008* #43). Most of the NBEs selected were from a previously characterized collection of embryonically active NB enhancers (*Manning et al., 2012*), with the remainder characterized as indicated in the Materials and Methods and *Supplementary file 1*. A total of 134 NBEs were used to make two libraries of transgenic fly lines, one expressing the $Cre_B$ fragment under the control of each of the 134 NBEs and the other similarly expressing the $Cre_C$ fragment. These lines thus collectively express $Cre_B$ and $Cre_C$ in a large number of distinct and often overlapping subsets of NBs (*Figure 1—figure supplement 1*). However, because the 134 enhancers are also typically active in mature neurons, the production of full-length Cre is not necessarily restricted to NBs (*Jenett et al., 2012*).

To ensure NB-specific reconstitution of Cre activity, we placed the $Cre_A$ and $Cre_{AB}$ fragments under the control of a compound enhancer formed by fusing individual enhancer elements of the NB-specific genes, *deadpan (dpn)* and *nervous fingers-1 (nerfin-1*; see Materials and methods). This synthetic *dpn-nerfin-1* enhancer (i.e. DNE) combines the complementary temporal characteristics of both component enhancers, maintaining strong, broad, and specific activity throughout embryonic neurogenesis (*Figure 1—figure supplement 2A,B*). Use of the DNE thus ensured that full-length, active Cre would be generated only in NBs where expression of the Cre fragments overlapped, and not in fully-differentiated neurons (*Figure 1G,I*). This enhancer also expresses in most of the NBs that give rise to the *Drosophila* CNS with the exception of those found in the late-developing optic lobes, and thus guarantees substantial coverage of the mature neurons found within the expression patterns of Gal4 lines.

To detect activity of the Split Cre constructs in vivo, we created transgenic flies carrying a reporter construct in which the floxed Gal80 gene, the expression of which is driven by a ubiquitously active Actin 5C promoter, is followed by the gene encoding the red fluorescent protein, tdTomato (*Figure 1J*). Expression of tdTomato from this construct, which we call Cre80Tom, thus identifies neurons in which Gal80 has been excised. Gal80 excision, identified by the appearance of tdTomato expression, is identifiable as early as embryonic stage eight when the $Cre_{AB}$ and $Cre_C$ fragments are driven by the DNE and it is widespread in the developing CNS by stage 13 (*Figure 1—figure supplement 2C*). The onset of Cre activity is sufficiently early to label many of the progeny of a defined NB lineage targetable by the R59E09-Gal4 line, previously identified by *Lacin and Truman*

*(2016)* (*Figure 1—figure supplement 2D*). These data suggest that Gal80 excision occurs relatively early.

The bipartite system, using the DNE-Cre$_{AB}$ and Cre$_C$ fragments expressed under the control of two different neuroblast enhancers (NBE$_{43H02}$ and NBE$_{44F03}$), also generated expression patterns in neuronal lineages of third instar larvae (*Figure 1K and L*). The expression patterns include not only the NBs in which Cre activity is reconstituted, but also the progeny of these NBs, since tdTomato expression is activated in all cells born within these lineages after Gal80 is excised. Although the expression patterns differ in the two cases, they share a small number of common NB lineages as is revealed by application of the tripartite Cre system using the NBE$_{44F03}$ and NBE$_{43H02}$ enhancers to drive Cre$_B$ and Cre$_C$, respectively, together with DNE-Cre$_A$ (*Figure 1M*). Expression in this case is limited to approximately three bilateral lineages in the ventral nerve cord (VNC) and two in the brain. These examples illustrate how the bi- and tripartite Split Cre constructs selectively reconstitute Cre activity in NBs targeted by individual NBEs, and demonstrate that the tripartite Split Cre system can be used to restrict Cre activity to only those NBs in which two distinct NBEs are active.

As this example illustrates, the tripartite system generates intersections of two NBE-Cre$_C$ expression patterns by substituting one Cre$_C$ fragment with a Cre$_B$ fragment driven by the identical NBE. To facilitate such substitutions, all NBE-Cre$_C$ insertions were made on Chromosome III and all NBE-Cre$_B$ insertions were made on Chromosome II. To evaluate the reproducibility of expression driven by individual NBEs, we examined NBE-Cre$_C$∩DNE-Cre$_{AB}$ >CreTom crosses for all 134 NBEs in multiple preparations (on average four per NBE) and compared these patterns with expression observed in all NBE-Cre$_B$∩DNE-Cre$_A$∩ DNE-Cre$_C$ >CreTom crosses. The large size of the patterns and the inability to reliably identify identical neurons and lineages across preparations prevented a systematic analysis, but in general the patterns were similar across preparations for any given cross (*Figure 1—figure supplement 3*) In most cases, patterns obtained with a given NBE-Cre$_B$ line also resembled the pattern obtained with the NBE-Cre$_C$ line made with the same enhancer (compare panels A and B in *Figure 1—figure supplement 3*). However, for 24 NBEs overt differences were observed (*Figure 1—figure supplement 3C* vs 3D). Apart from these NBEs, which have been marked with an asterisk in *Supplementary file 1* along with guidance as to their use, the tripartite system represents a reliable intersectional method for restricting Cre activity to subsets of NBs. The progeny of these NBs that are generated after Cre activation will not only express the reporter tdTomato, but will also fail to express the Gal80 transgene, thus permitting Gal4 to function.

## Using the bipartite and tripartite cre systems to restrict expression of TH-Gal4

The selective disinhibition of Gal4 activity in targeted lineages permits UAS-transgenes to be expressed in cells of those lineages whenever they lie within the expression pattern of a Gal4 driver. This allows targeted lineages to be parsed according to the properties of the mature neurons to which they give rise using cell-type specific Gal4 drivers. Such so-called 'cell class-lineage intersections' have been previously performed to identify subsets of neurons generated by Type II NBs of the *Drosophila* brain, which can be selectively targeted using a Type II-specific enhancer (*Ren et al., 2016*; *Ren et al., 2018*; *Ren et al., 2017*). Among the neurons generated by Type II NBs are several populations of dopaminergic neurons, identified by a Tyrosine Hydroxylase-specific Gal4 driver (TH-Gal4). Dopaminergic neurons are of considerable interest because of their roles in a variety of important neurobiological processes, including learning, sleep, and locomotion (for review see *Kasture et al., 2018*). The approximately 120–130 dopaminergic neurons in the *Drosophila* CNS are produced by diverse NBs and numerous reagents have been generated to selectively target them (*Aso et al., 2014*; *Friggi-Grelin et al., 2003*; *Xie et al., 2018*).

As a first test of the SpaRCLIn system, we therefore asked whether it could restrict expression of the TH-Gal4 driver (*Figure 2A*) to small numbers of distinct dopaminergic neurons based on their different lineages of origin. Using a small subset of the NBE-Cre$_C$ lines in combination with DNE-Cre$_{AB}$, we examined the expression patterns produced by intersection with TH-Gal4. The expression patterns produced by these intersections were noticeably reduced compared with the full pattern of the TH-Gal4 driver, but they typically still contained 10's of dopaminergic neurons distributed broadly across the neuraxis (*Figure 2B–C*). In cases where the expression patterns produced by the bipartite crosses shared a neuron (*Figure 2B–C*, arrows), combining the relevant NBEs using the tripartite system succeeded in isolating these neurons from most others in the two original crosses

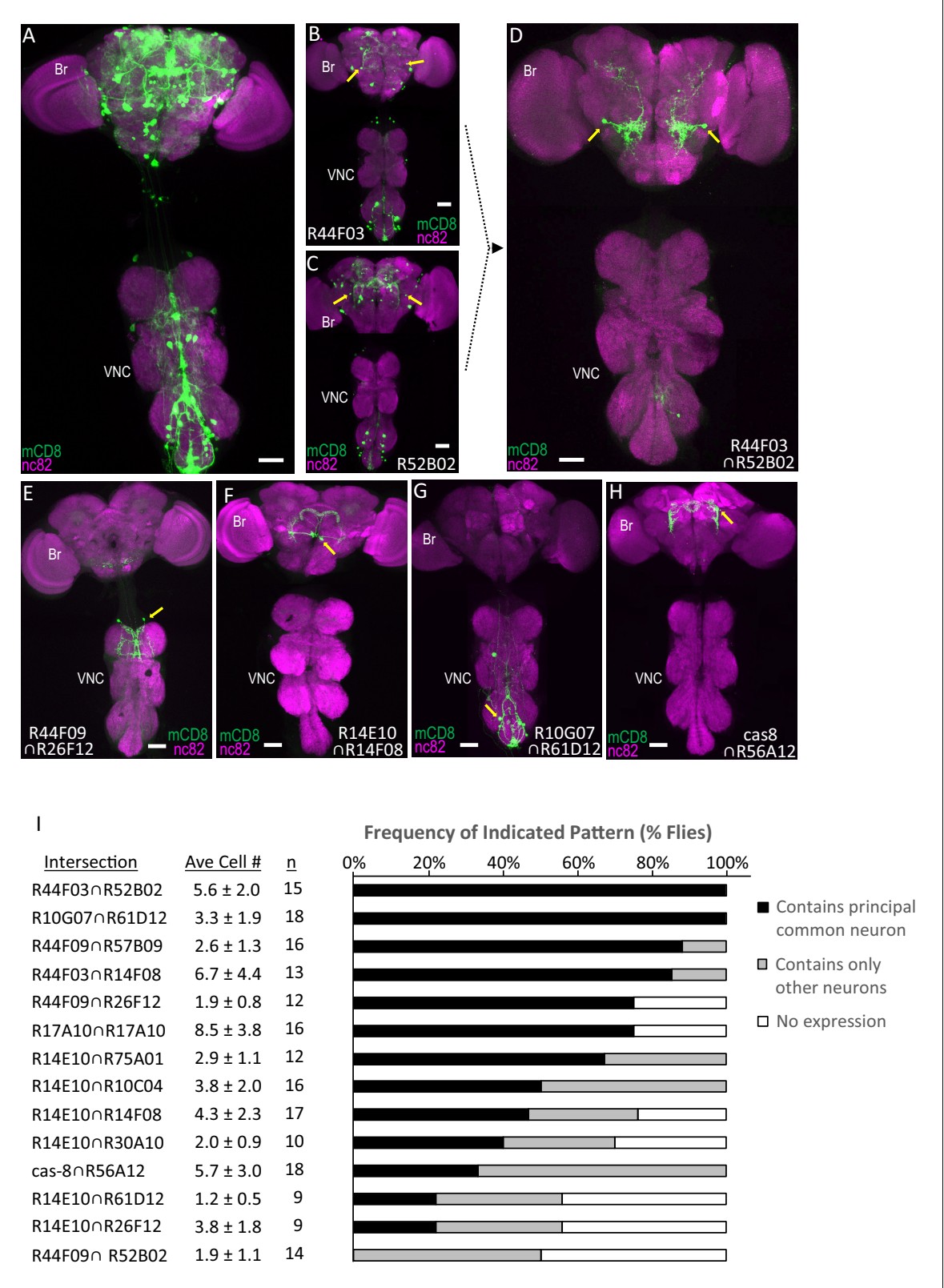

**Figure 2.** Parsing the TH-Gal4 expression pattern using SpaRCLIn. (**A**) Expression pattern of the TH-Gal4 driver revealed by UAS-mCD8GFP (green). In all panels: Anti-nc82 labeled neuropil (magenta); ventral nerve cord; VNC; brain; Br. (**B–D**) Restriction of TH-Gal4 expression using SpaRCLIn. (**B, C**) mCD8GFP expression (green) in mature dopaminergic neurons isolated using the bipartite SpaRCLIn system and two different NBEs (R44F03 and R52B02) to drive Cre_C expression. A neuronal pair common to both patterns is indicated (yellow arrows). (**D**) mCD8GFP expression (green) driven by the

*Figure 2 continued on next page*

*Figure 2 continued*

tripartite SpaRCLIn system at the intersection of the two NBE expression patterns in B and C. (**E–H**) Examples of TH-Gal4 restriction to small numbers of neurons using the tripartite system and the indicated pairs of NBEs. Scale bar: 50 µM. (**I**) Size and stereotypy of the restricted expression patterns produced by the indicated Step two intersections. The average number of neurons per preparation (± standard deviation) observed for each intersection is shown together with the number of preparations examined. For each, intersection the neuron that was most frequently observed across preparations (i.e. the 'principal common neuron') was identified and the percentage of preparations containing this neuron is shown in the bar graph (black bars) together with the percentage of preparations showing expression only in other neurons (gray bars) or no expression (white bars). Examples of principal common neurons are indicated by yellow arrows in D-H.

The online version of this article includes the following figure supplement(s) for figure 2:

**Figure supplement 1.** Stochastic expression within the TH-Gal4 pattern generated by SpaRCLIn.

**Figure supplement 2.** Reproducibility of labeling within the TH-Gal4 pattern generated by SpaRCLIn Expression patterns of all 16 CNS preparations for TH-Gal4$^{R14E10-CreB \cap R10C04-CreC}$.

(*Figure 2D*). In general, restricting NB expression using the tripartite system—by pairing the NBE-Cre$_C$ constructs with NBE-Cre$_B$ constructs made with different enhancers—produced significantly reduced expression patterns, sometimes consisting of one to two cells or bilateral cell pairs (*Figure 2D–H*).

The expression patterns from 14 NBE-Cre$_C$ ∩ NBE-Cre$_B$ intersections—produced by combining 15 distinct NBEs—were analyzed in detail to quantify both the average number of dopaminergic neurons and the stereotypy of expression for each intersection (*Figure 2I*). We found that the average number of labeled neurons per preparation did not exceed 8.5 (±3.8, n = 16) for any intersection and was less than 4.3 (±2.3, n = 17) for two-thirds of them. This sparseness of expression suggests that the NBEs tested do not overlap extensively in their NB expression patterns. Stereotypy of expression was also generally present despite considerable variability. Only in one extreme case, did there appear to be a complete absence of stereotypy, with all CNS preparations that had expression displaying a distinct pattern (*Figure 2—figure supplement 1*). For all other intersections, at least one principal neuron was found that was shared by multiple preparations, based on cell position and morphology (*Figure 2I*, black bars). For over half of the intersections, this principal common neuron was shared by 50% or more of preparations. In most cases, other neurons were also found, though preparations containing only such neurons typically occurred at lower frequency (*Figure 2I*, gray bars). Consistent with this variability of expression, neurons that recurred across preparations were not necessarily found in the same combinations (*Figure 2—figure supplement 2*).

The sparseness of labeling combined with the variability of expression likely accounts for why half of the intersections yielded at least one preparation without any expression. Interestingly, four of the seven intersections that yielded


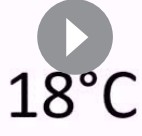

**Video 1.** Activation of neurons in the rk$^{pan}$-Gal4 pattern induces robust proboscis extension. rk$^{pan}$-Gal4 was used to drive expression of the heat-activated ion channel, UAS-dTrpA1. At 18˚C the channel is inactive and animals expressing it throughout the rk$^{pan}$-Gal4 pattern do not extend their proboscis. In contrast, at 31˚C when the channel is activated, animals display prolonged proboscis extension.

https://elifesciences.org/articles/53041#video1

preparations devoid of expression shared an enhancer (R14E10), suggesting that particular enhancers may strongly influence the extent of labeling. Variability of labeling also appeared to be enhancer-dependent in that use of the same enhancer (i.e. R17A10) to drive both $Cre_B$ and $Cre_C$ components did not necessarily reduce stochasticity. Indeed, although all preparations that had expression shared a common identifiable neuron in this case (*Figure 2I*), their expression in other neurons varied considerably. A possible source of this variability of expression is weak NBE activity that results in lowered expression of Cre components and consequently more sporadic reconstitution of Cre activity. More work will be required to examine this hypothesis. Regardless, our results demonstrate SpaRCLIn's ability to substantially restrict expression of a Gal4 driver with sufficient stereotypy in single neurons to be useful for the neuronal manipulations employed in neural circuit mapping.

## Functional circuit-mapping using SpaRCLIn

To examine SpaRCLIn's efficacy for circuit mapping, we used it to identify neural substrates of proboscis extension (PE), a motor pattern normally elicited by gustatory stimuli, but also by the hormone Bursicon in newly eclosed flies (*Peabody et al., 2009*). Robust PE can be readily induced even in older flies using a driver (rk$^{pan}$-Gal4) that selectively expresses in Bursicon-responsive neurons (*Video 1*, *Figure 3A,B*; *Diao and White, 2012*). Expressing the heat-sensitive ion channel UAS-dTrpA1 under the control of this driver, we performed an initial ('Step 1') screen of the $Cre_C$ library using the bipartite SpaRCLIn system (*Figure 3—figure supplement 1A*). In this screen, crosses were conducted between each NBE-$Cre_C$ line and a line that combined all other components, including Cre80Tom, DNE-$Cre_{AB}$, rk$^{pan}$-Gal4, and UAS-dTrpA1. To facilitate visualization of neurons within the resulting expression pattern without requiring additional genomic insertions, we used a dual expression construct (Cre80Tom- GFP) that contained actin̂Gal80̂myr-tdTomato and a 10XUAS-mCD8GFP reporter (*Figure 3—figure supplement 2*). Progeny were videorecorded in small chambers on a temperature-controlled plate and assayed for heat-induced PE. Interestingly, several different PE phenotypes were apparent, but only those that involved full extension of the proboscis could be reliably scored under our assay conditions and we therefore focused on the latter. Applying this criterion, we identified 23 NBE-$Cre_C$∩DNE-$Cre_{AB}$ intersections for which UAS-dTrpA1 activation reliably induced robust PE in greater than 50% of the progeny. The expression patterns resulting from these $Cre_{AB∩C}$∩rk$^{pan}$-Gal4 (i.e. Step 1) intersections, examined using a UAS-GFP reporter, were clearly restricted relative to rk$^{pan}$-Gal4 expression (*Figure 3C–D*), but they were insufficiently sparse to readily identify the neurons—or population of neurons—responsible for inducing the PE motor pattern.

Taking advantage of SpaRCLIn's ability to further restrict expression, we used the tripartite system to carry out a second ('Step 2') screen in which the 23 identified NBE-$Cre_C$ components were combined pairwise with NBE-$Cre_B$ components made using the same 23 enhancers (*Figure 3—figure supplement 1B*). The latter were selected from the NBE-$Cre_B$ library and crosses were made that combined distinct NBE-$Cre_B$ and NBE-$Cre_C$ components with DNE-$Cre_A$, rk$^{pan}$-Gal4, and Cre80-Tom-GFP. These Step two crosses resulted in $Cre_{A∩B∩C}$∩rk$^{pan}$-Gal4 intersections that were assayed for PE as before. Approximately 70 intersections were tested before screening was discontinued because 11 intersections had already yielded PE phenotypes in greater than 50% of flies. The phenotype observed was typically less sustained than that produced by activation of the full rk$^{pan}$-Gal4 expression pattern in that activation typically caused rhythmic, rather than tonic, extension of the proboscis, which after prolonged heating often transitioned to lifting of the rostrum rather than full extension (*Video 2*; *Figure 3E*).

The rk$^{pan}$-Gal4 expression patterns in flies exhibiting this phenotype were substantially reduced for many of the intersections tested and they consistently included particular neurons in the subesophageal zone (SEZ) that were characterized by somata near the saddle, broad arbors along the superior gnathal ganglion (GNG), and axons that extended medially before turning, with one branch coursing down each side of the midline and then turning laterally along the medial-inferior edges of the GNG (*Video 3*). Two closely apposed neurons of this type were observed, sometimes as bilateral pairs (*Figure 3F*), and sometimes on only one side (*Figure 3G*). These neurons, which we call the PE$^{rk}$ neurons, were notably prominent in the 16H11-$Cre_B$∩44F09-$Cre_C$ intersection, where they constituted the entire expression pattern of 16 animals (n = 78 total), all of which exhibited PE induction upon heating. Indeed, all 36 animals from this intersection that tested positive for the PE phenotype

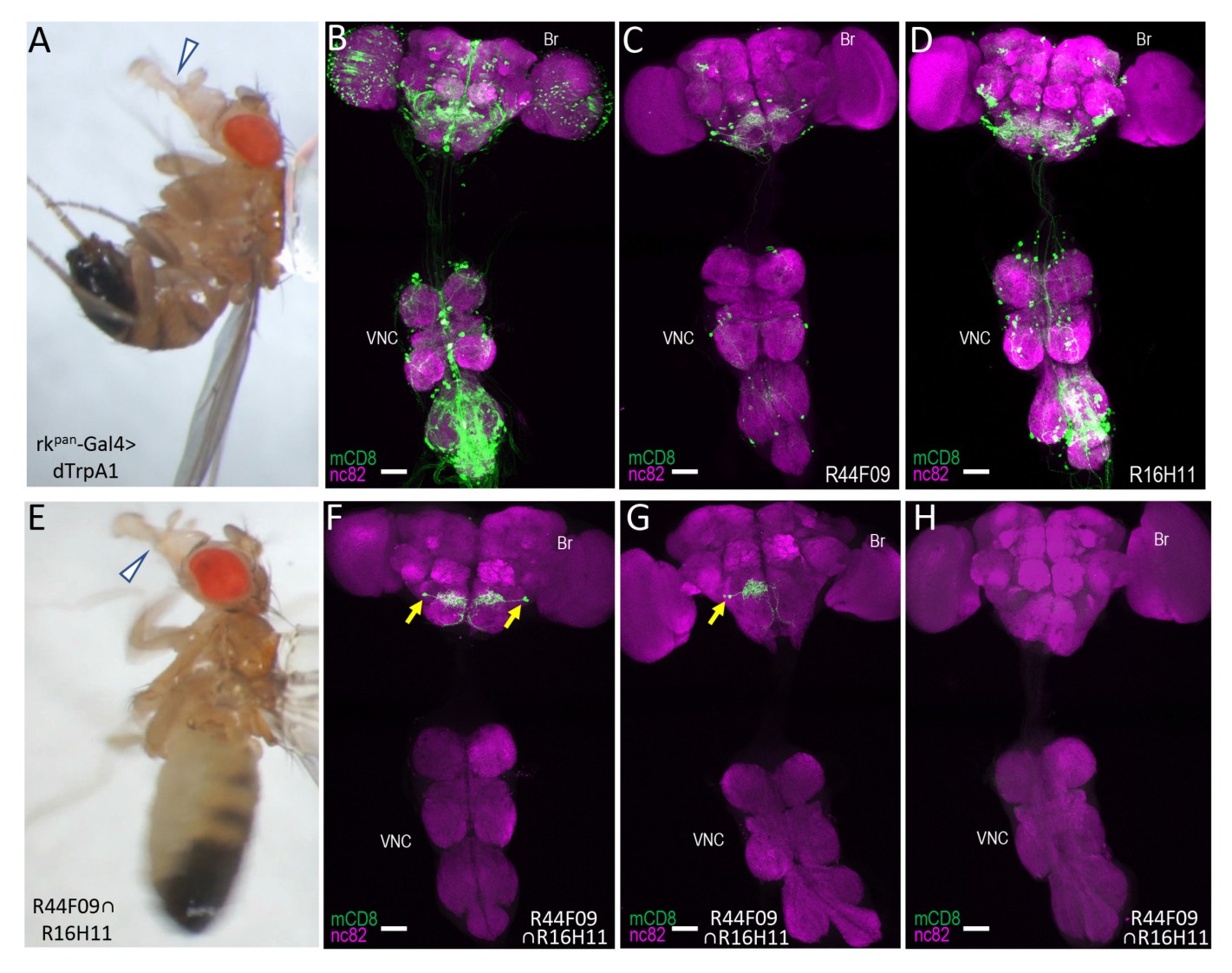

**Figure 3.** Identification of command neurons for PE within the rk^pan- Gal4 pattern. (**A**) Induced PE (arrowhead) in a fly expressing the heat-sensitive ion channel dTrpA1 under the control of the rk^pan-Gal4 driver. Labels as described in the legend of *Figure 2A*. (**B**) Expression pattern of the rk^pan-Gal4 driver revealed by UAS-mCD8GFP (green). In all panels: Anti-nc82 labeled neuropil (magenta); ventral nerve cord: VNC; brain: Br. (**C–D**) mCD8GFP expression (green) in mature subsets of RK-expressing neurons isolated using the bipartite SpaRCLIn system and NBEs R44F09 and R516H11 to drive Cre_C expression. (**E**) PE induced in a fly expressing dTrpA1 in the PE^rk neurons, isolated using the tripartite system with the R44F09 and R16H11 NBEs to parse the rk^pan-Gal4 pattern. (**F–H**) Typical expression patterns in rk^pan-Gal4^R44F09 ∩ R16H11 flies, showing expression in both bilateral pairs of PE^rk neurons (**F**), one neuron of each of the two bilateral pairs of PE^rk neurons (**G**), or no neurons (**H**). All scale bars: 50 μM.

The online version of this article includes the following figure supplement(s) for figure 3:

**Figure supplement 1.** Workflow for SpaRCLIn Screens.

**Figure supplement 2.** Dicistronic vector with floxed Gal80 and UAS constructs.

**Figure supplement 3.** rk^R16H11-CreB∩R25G06-CreC-Gal4 expression patterns include PE^rk and other neurons.

and were successfully dissected showed expression in the PE^rk neurons, while none of the animals (n = 38) that tested negative had such expression (*Figure 3I*). Most of the latter, in fact, had little to no expression. Similar results were obtained with a second intersection (44F09-Cre_B∩10G07-Cre_C). All 19 animals that exhibited induced PE in this intersection had expression in the PE^rk neurons, and in three animals these were the only neurons present. A third intersection that yielded the PE pheno-type in all animals likewise showed consistent expression in the PE^rk neurons, but the correlation between the PE phenotype and expression in these neurons was somewhat less readily established

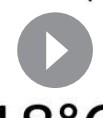

rk$^{pan}$-Gal4$^{R44F09Cre_B \cap R16H11Cre_C}$

>UAS-dTrpA1

18°C

**Video 2.** Activation of the PE$^{rk}$ neurons induces robust, rhythmic proboscis extension. The tripartite SpaRCLIn system isolates a subset of neurons within the rk$^{pan}$-Gal4$^{DNE-CreA \cap R16H11-CreB \cap R44F09-CreC}$ intersection called the PE$^{rk}$ neurons. When activated using dTrpA1 and a temperature of 31°C repeated, rhythmic proboscis extension is induced.

https://elifesciences.org/articles/53041#video2

(embryonic) neuroblasts (*Figure 4B*). Cre$_{AB}$ will thus be available to reconstitute Cre activity only with complimentary Cre$_C$ fragments that are also expressed at this time. Cre$_C$s whose expression is driven by NBEs that become active only after the elimination of Cre$_{AB}$ from neuroblasts, will not lead to the generation of Gal4-competent neurons. Expression patterns resulting from the combination of FRTerminator with NBE-Cre$_C$s will thus, in general, be reduced relative to those produced by DNE-Cre$_{AB}$ (*Figure 4C,D*).

To determine whether the FRTerminator might therefore expedite parsing of Gal4 expression using the SpaRCLIn system, we repeated selected crosses from the rk$^{pan}$-Gal4 Step one screen described above. We focused on the 23 NBE-Cre$_C$ lines that yielded flies with PE phenotypes, combining each with the

because of expression in other neurons (5.6 ± 1.8; n = 14 preparations; *Figure 3—figure supplement 3*).

## FRTerminator: a self-excising DNE-Cre$_{AB}$ to facilitate fine-mapping in Step one screens

The above examples demonstrate that SpaRCLIn can be used to rationally parse the expression patterns of Gal4 drivers using the workflow shown in *Figure 3—figure supplement 1*. One challenge to using this system, however, is the large number of transgenes required to implement it. This is especially true for Step two screening with the tripartite system. To mitigate this burden, we have created several reagents that will facilitate use of the system. In addition to the Cre80Tom-GFP construct described above, we have developed other dicistronic constructs to facilitate manipulating neuronal activity in SpaRCLIn screens (see Key Resources Table). These include constructs and fly lines for Cre80-Kir2.1 and Cre80-dTrpA1 (*Figure 3—figure supplement 2*). In addition, we have developed an alternate Step one strategy that may avert the need for Step two screening in favorable cases.

The alternate strategy uses a transiently expressed DNE-Cre$_{AB}$ designed to be active only during early stages of neurogenesis. This construct, which we call 'FRTerminator,' is self-excising in that it is flanked by Flp Recombination Target (FRT) sites and encodes a Flp recombinase gene that is co-expressed with Cre$_{AB}$ (*Figure 4A*). Upon expression under control of the DNE enhancer, this construct will remove the Cre$_{AB}$ gene and thus limit its expression to early

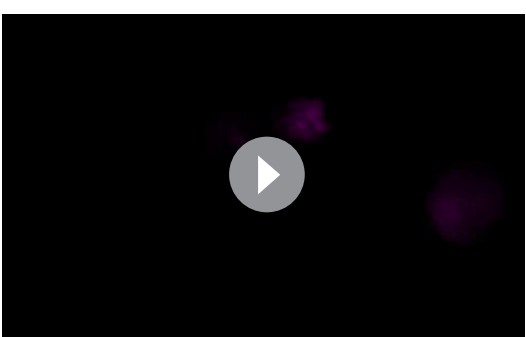

**Video 3.** Neuroanatomical location and projection pattern of the PE$^{rk}$ neurons. GFP-labeled PE$^{rk}$ neurons (green) were imaged by confocal microscopy to show the location of their somata and their arborization. Neuropil labeled by nc82 antibody is shown in blue to identify brain regions.

https://elifesciences.org/articles/53041#video3

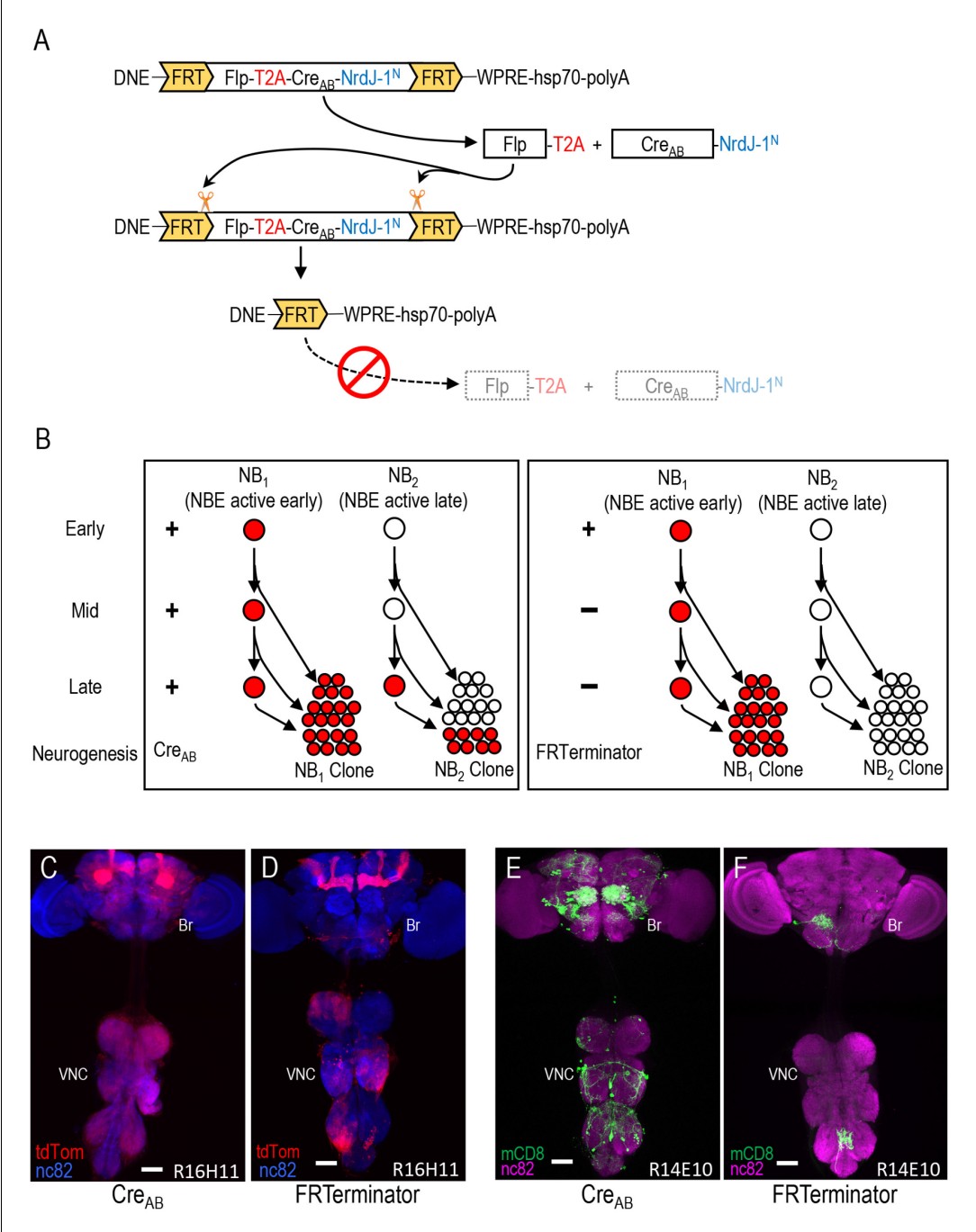

**Figure 4.** Limiting Cre activity to early NBs using FRTerminator. (**A**) The FRTerminator construct: a DNE-$Cre_{AB}$ that terminates its own expression. The FRTerminator expression cassette contains sequences for the Flp recombinase and $Cre_{AB}$-NrdJ-$1^N$ linked by a viral T2A sequence to ensure separate translation of the two gene products. The entire cassette is flanked by FRT sites. Upon expression of the cassette—which will occur in NBs at the onset of neurogenesis—Flp will excise the cassette, thus terminating any further expression of both Flp and $Cre_{AB}$-NrdJ-$1^N$. (**B**) Schematic comparing the consequences of DNE-$Cre_{AB}$ (left box) and FRTerminator (right box) action in two NB lineages ($NB_1$ and $NB_2$) in which an NBE (used to drive expression of $Cre_C$) is active. In $NB_1$ the NBE is active early in neurogenesis and $Cre_C$ will therefore be expressed in the young neuroblast. In contrast, the NBE becomes active only late in neurogenesis in $NB_2$ and $Cre_C$ is therefore only present in the older NB. Because DNE-$Cre_{AB}$ is expressed throughout neurogenesis, it will be available to reconstitute full-length Cre whenever $Cre_C$ is expressed. This means that Gal80 will be excised and tdTomato expression turned on (red) early in $NB_1$—leading to the labeling of all progeny—and late in $NB_2$—leading to labeling of only late-generated progeny. In contrast, FRTerminator is present only early in neurogenesis and Cre reconstitution (and tdTomato expression) will therefore occur only in $NB_1$. No progeny of the $NB_2$ clone will be labeled and the overall pattern of labeling will thus be diminished. (**C–D**) NB lineages targeted using $NBE_{R16H11}$-$Cre_C$ and either the DNE-$Cre_{AB}$ construct of the bipartite SpaRCLIn system (**C**), or FRTerminator (**D**). NB progeny are visualized with tdTomato (red) after

*Figure 4 continued*

excision of Gal80 by Cre. The breadth of tdTomato expression when using DNE-Cre$_{AB}$ compared with FRTerminator reflects the loss of sublineages generated by NBs in which the R16H11 enhancer becomes active only later in neurogenesis, as illustrated in B. Anti-nc82 labeled neuropil (blue); ventral nerve cord; VNC; brain; Br. Scale bar: 50 µM. (E–F) Restriction of the rk$^{pan}$-Gal4 expression pattern by SpaRCLIn using R14E10-Cre$_C$ with DNE-Cre$_{AB}$ (E) or FRTerminator (F). FRTerminator significantly reduces the expression pattern compared with the restriction obtained with DNE-Cre$_{AB}$, labeling principally the PE$^{rk}$ neurons. Reporter: UAS-mCD8GFP (green); Anti-nc82 labeled neuropil (magenta); ventral nerve cord; VNC; brain; Br. Scale bar: 50 µM.

FRTerminator, rk$^{pan}$-Gal4 and Cre80-GFP. Progeny were tested for PE upon dTrpA1 activation. We found that three NBE-Cre$_C$ lines (44F09, 57B09, and 14E10) produced progeny with PE phenotypes at frequencies ranging from 9–17%. Although these frequencies were considerably lower than those obtained using DNE-Cre$_{AB}$, the resulting expression patterns were substantially sparser compared with those of progeny from DNE-Cre$_{AB}$ crosses (*Figure 4E,F*). All animals examined that had PE phenotypes also included in their expression patterns the PE$^{rk}$ neurons (n = 40). In contrast, only one of the animals examined that lacked the phenotype had these neurons (n = 39). A strong correlation between PE and the presence of the PE$^{rk}$ neurons was thus observed, again permitting the conclusion that these neurons are substrates for the behavioral phenotype. We conclude that FRTerminator-based Step one screens may serve as a useful shortcut to serial Step one and Step two screens for restricting Gal4 expression and identifying functionally important neuronal subsets.

## Discussion

The SpaRCLIn system introduced here permits the refined targeting of neurons within a group of interest based on both their developmental origins and their patterns of gene expression in the terminally differentiated state. By permitting the combinatorial targeting of many, if not most, of the neuroblasts that generate the mature CNS, the SpaRCLIn system provides end-users with a comprehensive, 'off-the-shelf' set of reagents for systematically isolating and characterizing the anatomy and function of specific neurons. The reagents that we have created include extensive lineage-selective Split Cre lines for bipartite (Step 1) and tripartite (Step 2) neuronal screens, in addition to a range of tools that facilitate application of the system. Dual effector and reporter constructs reduce the number of transgenes required to implement the system, and a self-terminating Split Cre component (i.e. FRTerminator) can be used to expedite screening in favorable circumstances. The system is compatible with existing Gal4 driver lines and the examples provided here indicate that it is capable of routinely parsing Gal4 expression patterns into subsets of neurons numbering in the single digits.

### Utility of SpaRCLIn to circuit mapping

Our use of SpaRCLIn to identify the RK-expressing neurons that trigger robust proboscis extension demonstrates SpaRCLIn's ability to systematically parse a neuronal group and identify the functionally relevant subset. Just over 200 crosses—134 crosses for the Step 1 screen of NBE-Cre$_C$ lines and 70 NBE-Cre$_{B∩C}$ Step two crosses—were required to identify two pairs of command-like neurons capable of inducing PE upon activation (i.e. the PE$^{rk}$ neurons). Importantly, we discontinued the Step two screen after testing 70 of the 253 possible intersections because of evident redundancy of the command-like neurons. The latter were prominent in the expression patterns of numerous independent Step two intersections and were readily correlated with PE induction in three that produced particularly reduced expression patterns. In the intersection with the sparsest expression, the two pairs of PE-inducing neurons often comprised the entire observable pattern in flies that had the PE phenotype, illustrating the extreme reduction in expression achievable with SpaRCLIn. The demonstration that the PE$^{rk}$ neurons can be isolated in single crosses using the FRTerminator indicates that this reduction in expression can be attained without the labor of Step two screening. However, the lower frequency of the PE phenotype in FRTerminator crosses in our example also suggests that FRTerminator-based screens may require testing more animals for each intersection than a standard Step one screen in order to reliably identify positives.

Activation of the PE$^{rk}$ neurons elicits rhythmic proboscis extension, rather than the tonic PE elicited by activation of all rk$^{pan}$-Gal4 neurons. This suggests that additional RK-expressing neurons—

perhaps lacking command capability—modulate the effects of activating the PE[rk] neurons. Based on their induction of rhythmic extension and their apparent lack of a projection to the proboscis muscles, we conjecture that the PE[rk] neurons identified here are not motor neurons, the activation of which results in tonic and often partial PE (*Gordon and Scott, 2009*; *Schwarz et al., 2017*). Similarly, the anatomy of the PE[rk] neurons differs from that of other identified neurons that can drive PE when activated, including second-order projection neurons (*Kain and Dahanukar, 2015*), modulatory neurons (*Marella et al., 2012*), and a local SEZ interneuron called the Fdg-neuron (*Flood et al., 2013*). Like the Fdg-neuron, however, the neurons identified here seem to function in a premotor capacity, perhaps as part of the central pattern generator for PE that regulates fly feeding (*Itskov et al., 2014*). Further work will be required to determine the precise role of the PE[rk] neurons in the feeding circuitry and their relationship to other identified neurons involved in PE.

It also remains to be determined whether activation of both PE[rk] neurons is required to induce the PE phenotype. Indeed, from the standpoint of the efficacy of the SpaRCLIn system it is important to ask why SpaRCLIn failed to separate these two pairs of neurons. The similarity of the two PE[rk] neurons in both soma position and projection pattern is consistent with their being part of the same lineage. Such neurons will necessarily be more difficult to parse using SpaRCLIn, which can separate neurons within the same lineage only based on their birth order. What would be required to do so is having two NBEs that are active in the same lineage but at different times so that they separate earlier- from later-born neurons. Such NBEs, by generating Cre only in older neuroblasts, will generate sublineages of Gal4-competent neurons. Although many of the NBE's used to make our Cre$_B$ and Cre$_C$ libraries clearly generate such sublineages—based on the patterns shown in *Figure 1—figure supplement 1*—it is doubtful that they cover more than a fraction of all temporal windows of neurogenesis in all neuronal lineages. A method for systematically isolating sublineages of later born neurons using SpaRCLIn may become possible if neuroblast-specific enhancers can be found that are selectively active at later stages of neurogenesis. These could then be used in lieu of the DNE used here. Candidates for such enhancers are those that determine expression of the so-called 'temporal transcription factors' that regulate the progressive divisions of many neuroblasts (*Doe, 2017*).

## Variability of SpaRCLIN expression

Although stochasticity is not an uncommon feature of many expression systems (*Bohm et al., 2010*; *Tastekin and Louis, 2017*), the variability of expression generated by SpaRCLIn was notable. Even for intersections that reliably produce very similar expression across animals, it is not common to get exactly the same pattern twice. The infidelity of expression may derive, at least in part, from intrinsic stochasticity of NBE activity. However, our results indicate that individual NBEs drive expression in broadly reproducible patterns. Other factors that may contribute to expression variability include the strength and/or temporal properties of NBE activity. If a Cre component is only weakly expressed, or expressed late during neurogenesis, a limited amount of active, full-length Cre may be produced and excision of Gal80 may be sporadic in the expressing neuroblasts. Also, the success of Cre reconstitution may vary, and may be particularly low when the religation of three fragments is required. Although the two split inteins used in the SpaRCLIn system were chosen based on their favorable reaction kinetics as determined in vitro, the speed, efficiency, and variability with which they react in different types of cells remain unknown. Finally, because the efficacy of Cre-mediated excision depends on the distance between the loxP sites flanking the excised fragment, it may prove possible to increase the efficacy of excision—and thus reduce variability of expression—using strategies that decrease the distance between the loxP sites flanking the fragment to be excised.

While further work will be required to identify the sources of variable expression within the system, the observed stochasticity is not necessarily a disadvantage for circuit-mapping applications, as illustrated by the examples presented here. By providing partially 'randomized' expression patterns, SpaRCLIn permits causative relationships to be inferred between groups of manipulated neurons and the effects produced by their manipulation (*Jazayeri and Afraz, 2017*). Such randomization has been commonly exploited in so-called 'Flp-out' methods that rely on stochastically induced recombinase activity to remove an FRT-flanked gene or transcription stop cassette (*Flood et al., 2013*; *Gordon and Scott, 2009*; *Kain and Dahanukar, 2015*). This logic is naturally implemented in SpaRCLIn, but because randomness of expression is considerably more constrained than that observed in systems that rely on strictly stochastic methods, and because the size of the expression patterns is typically small, correlations can be readily established.

One consequence of SpaRCLIn's stochasticity that must be considered in circuit mapping applications, however, is the lowered frequency of bilateral labeling. Most neurons occur as members of bilateral pairs and we observed numerous instances in which SpaRCLIn-derived expression patterns contained only a single member of each pair in a given preparation—presumably due to the variable success of Gal80 excision in both NBs giving rise to the pair. The reduced bilateral representation of neurons may likewise reduce the frequency of phenotypes observed as a consequence of a particular manipulation if, for example, both neurons in a pair must be affected to produce a phenotype. This is often the case for suppression of function, where both neurons in the pair must be inhibited. It is therefore possible that SpaRCLIn will be most effective in applications that involve neuronal activation where unilateral manipulations are often sufficient to generate an effect as they are for proboscis extension.

## Other considerations in the use of SpaRCLIn

The ability of SpaRCLIn to isolate a given set of neurons of interest in a Gal4 pattern depends critically on the extent to which the various Split Cre components are expressed in the neuroblast lineages of the fly. This will be determined both by the breadth of NB expression of the DNE enhancer used here to delimit Cre activity and by the collective coverage of NB expression provided by the NBEs represented in the libraries of $Cre_B$ and $Cre_C$ lines. Our analysis of $3^{rd}$ instar larval expression in DNE∩NBE intersections (*Figure 1—figure supplement 1* and data not shown) indicates that many, if not most, NB lineages of the ventral nerve cord and central brain are likely represented within the libraries. Indeed, many lineages are clearly represented multiple times in that different intersections repeatedly isolated the same neurons (e.g. the $PE^{rk}$ neurons) for both the $rk^{pan}$-Gal4 and TH-Gal4 drivers. It is less clear, however, that all members of each lineage are represented as not all NBE's are active during early NB divisions. This is evident from the restriction in NB expression observed when the FRTerminator construct is used, since this construct acts by eliminating lineages or sublineages in which Cre activity is initiated sometime after neurogenesis has begun. It is also clear that the DNE does not express efficiently in NB lineages in the optic lobe (data not shown). To extend the capability of the system to include these lineages will require either the development of a more general neuroblast-specific enhancer or augmenting the system to include an enhancer that specifically targets optic lobe NBs.

The effectiveness of SpaRCLIn also depends critically on the success of Cre reconstitution by the system, which is effected by two pairs of split inteins (*Shah and Muir, 2011*; *Shah and Muir, 2014*). These trans-splicing protein fragments function naturally in protein religation and are an emerging technology for use in transgenic animals (*Hermann et al., 2014*; *Wang et al., 2018*; *Wang et al., 2012*). Their advantages are that they lend themselves readily to intersectional methods, are genetically encoded, and in numerous cases display rapid reaction kinetics and low cross-reactivity. A disadvantage, on which some recent progress has been made (*Stevens et al., 2017*), is that most split inteins require specific flanking amino acid residues in the proteins to which they are fused, in particular a cysteine or serine residue immediately downstream of the N-intein. We were able to create self-ligating split Cre fragments capable of reconstituting full-length, active Cre enzyme in *Drosophila* NBs by choosing breakpoints in the Cre sequence preceded by a serine residue—the native condition of the NrdJ-1 and gp41-1 split inteins used here (*Carvajal-Vallejos et al., 2012*). Orthogonal (i.e. non-interacting) split inteins thus represent attractive tools for reconstituting the function of multiply split proteins, a methodology that should be applicable in other model organisms.

## Conclusions and future development

Although sophisticated methods for neuronal targeting have been a hallmark of neurobiological studies on the fly, and single cell manipulations are being leveraged in a growing number of cases to elucidate *Drosophila* brain circuits, targeting every cell in the fly CNS remains an aspirational goal. Recent progress towards this goal has been made using the Split Gal4 system (*Dionne et al., 2018*; *Tirian et al., 2017*), and innovative methods continue to be developed using emerging tools (*Garcia-Marques et al., 2019*). An advantage of SpaRCLIn is that it represents a relatively small set of stand-alone reagents for high-specificity neuronal targeting that can be used with the many existing components of the Gal4-UAS system. Importantly, SpaRCLIn also represents an open resource that can readily be augmented by end-users. As methods improve for rationally identifying NB lineages

based on gene expression and enhancer activity, the existing SpaRCLIn libraries can be supplemented with lines that together permit the selective targeting of an increasing number of neuroblast lineages. The NB-specific enhancers recently identified by *Lacin and Truman (2016)* and used here to characterize our split Cre components (*Figure 1—figure supplement 2D*) provide good examples of reagents that can be used to improve the SpaRCLIn libraries. By combining these libraries with an optimized set of Gal4 drivers that express in distinct subsets of brain cells (distinguished, for example, by transcription factor expression), one can imagine having a set of 3 libraries that in combination can selectively target most neurons in CNS.

# Materials and methods

## Key resources table

| Reagent type (species) or resource | Designation | Source or reference | Identifiers | Additional information |
|---|---|---|---|---|
| Genetic reagent (*D. melanogaster*) | w; Sco/Cyo; Rk$^{pan}$-Gal4 | *Diao and White, 2012* | N/A | |
| Genetic reagent (*D. melanogaster*) | w; UAS-TrpA1(attP16);+ | Bloomington *Drosophila* Stock Center | RRID:BDSC_26263 | |
| Genetic reagent (*D. melanogaster*) | w; +; TH-Gal4 | J Hirsh and S Birman | THGal4-1 | |
| Genetic reagent (*D. melanogaster*) | yw; mCD8-GFP/Cyo;+ | Gift of Liqun Luo | N/A | |
| Genetic reagent (*D. melanogaster*) | w; DNE-Cre$_{AB}$ (JK22C);+ | This paper | HJ210 | *Supplementary file 4* |
| Genetic reagent (*D. melanogaster*) | w, DNE-Cre$_B$[su(Hw)attP8]; Pin/CyO; TM3,Sb1/ TM6B,Hu1,Tb1 | This paper | HJ201 | *Supplementary file 4* |
| Genetic reagent (*D. melanogaster*) | w; +; DNE-Cre$_A$ (VK00027) | This paper | HJ200 | *Supplementary file 4* |
| Genetic reagent (*D. melanogaster*) | w; DNE-Cre$_A$ (attP40); + | This paper | HJ200 | *Supplementary file 4* |
| Genetic reagent (*D. melanogaster*) | w, Cre80Tom [su(Hw)attP8]; Pin/CyO; TM3,Sb$^1$/ TM6B,Hu$^1$,Tb$^1$ | This paper | HJ223 | *Supplementary file 4* |
| Genetic reagent (*D. melanogaster*) | w, Cre80Tom-GFP [su(Hw)attP8]; Pin/CyO; TM3,Sb$^1$/ TM6B,Hu$^1$,Tb$^1$ | This paper | HJ224 | *Supplementary file 4* |
| Genetic reagent (*D. melanogaster*) | w; DNE-Frterminator (attP40);+ | This paper | HJ473 | *Supplementary file 4* |
| Genetic reagent (*D. melanogaster*) | w; +; DNE-Cre$_{BC}$ (attP2) | This paper | HJ209 | *Supplementary file 4* |
| Genetic reagent (*D. melanogaster*) | UAS-Cre$_C$ (attP40) | This paper | HJ226 | *Supplementary file 4* |
| Genetic reagent (*D. melanogaster*) | CreStop (attP2) | This paper | HJ225 | *Supplementary file 4* |
| Genetic reagent (*D. melanogaster*) | w, Cre80-Kir2.1 [su(Hw)attP8]; Pin/CyO; TM3,Sb$^1$/TM6B,Hu$^1$,Tb$^1$ | This paper | HJ472-actin5C-FloxSyn 21Gal80-pMUH-Kir2.1 | *Supplementary file 4* |
| Genetic reagent (*D. melanogaster*) | w, Cre80-TrpA1 [su(Hw)attP8]; Pin/CyO; TM3,Sb$^1$/ TM6B,Hu$^1$,Tb$^1$ | This paper | HJ382-actin5C-FloxSyn 21Gal80-10XUAS-dTrpA1 | *Supplementary file 4* |
| Genetic reagent (*D. melanogaster*) | R59E09-Gal4(attP2) | Bloomington *Drosophila* Stock Center | RRID:BDSC_39220 | |

*Continued on next page*

*Continued*

| Reagent type (species) or resource | Designation | Source or reference | Identifiers | Additional information |
|---|---|---|---|---|
| Antibody | Rat monoclonal anti-Mouse CD8a | Invitrogen Life Technologies | RRID:AB_10392843 | 1:100 dilution |
| Antibody | Goat polyclonal Alexa Fluor 488 anti-rat IgG (H+L) | Invitrogen Life Technologies | RRID:AB_2534074 | 1:500 dilution |
| Antibody | Rabbit polyclonal Anti-RFP Antibody | Rockland, Imm. | RRID:AB_2209751 | 1:5000 dilution |
| Antibody | Mouse monoclonal Anti-futsch Antibody | Developmental Studies Hybridoma Bank | RRID:AB_528403 | 1:50 dilution |
| Antibody | Rabbit polyclonal Living Colors DsRed Polyclonal Antibody | CLONTECH Laboratories | RRID:AB_10013483 | 1:500 dilution |
| Antibody | Goat polyclonal Cy3 AffiniPure Goat Anti-Rabbit IgG (H+L) | Jackson Immuno-Research | RRID:AB_2338006 | 1:500 dilution |
| Antibody | Mouse monoclonal nc82 | Developmental Studies Hybridoma Bank | RRID:AB_2314866 | 1:50 dilution |
| Antibody | Goat polyclonal Alexa Fluor 488 goat anti-mouse IgG (H+L) | Invitrogen Life Technologies | RRID:AB_2534069 | 1:500 dilution |
| Antibody | Goat polyclonal Alexa Fluor 647-AffiniPure Goat Anti-Mouse IgG (H+L) | Jackson Immuno-Research | RRID:AB_2338914 | 1:500 dilution |
| Recombinant DNA reagent | Cre$_A$ | This paper | HJP-176-IVS-Syn21-Cre$_A$-gp41-1$^N$-WPREw | For making CreA line from promoter |
| Recombinant DNA reagent | Cre$_B$ | This paper | HJP-180-IVS-Syn21-gp41-1$^C$-Cre$_B$-NrdJ-1$^N$-WPREw | For making CreB line from promoter |
| Recombinant DNA reagent | Cre$_C$ | This paper | HJP-179-IVS-Syn21-NrdJ-1$^C$-Cre$_C$-WPREw | For making CreC line from promoter |
| Recombinant DNA reagent | Cre$_{AB}$ | This paper | HJP-178-IVS-Syn21-Cre$_{AB}$-NrdJ-1$^N$-WPREw | For making CreAB line from promoter |
| Recombinant DNA reagent | Cre$_{BC}$ | This paper | HJP-177-IVS-Syn21-gp41-1$^C$-Cre$_{BC}$-WPREw | For making CreBC line from promoter |
| Recombinant DNA reagent | Cre$_A$U | This paper | HJP-194-IVS-Syn21-Cre$_A$-gp41-1$^N$-WPREUw | For making CreA from enhancer |
| Recombinant DNA reagent | Cre$_B$U | This paper | HJP-195-IVS-Syn21-gp41-1$^C$-Cre$_B$-NrdJ-1$^N$-WPREUw | For making CreB line from enhancer |
| Recombinant DNA reagent | Cre$_C$U | This paper | HJP-196-IVS-Syn21-NrdJ-1$^C$-Cre$_C$-WPREUw | For making CreC line from enhancer |
| Recombinant DNA reagent | Cre$_{AB}$U | This paper | HJP-208-IVS-Syn21-Cre$_{AB}$-NrdJ-1$^N$-WPREUw | For making CreAB line from enhancer |
| Recombinant DNA reagent | Cre$_{BC}$U | This paper | HJP-207-IVS-Syn21-gp41-1$^C$-Cre$_{BC}$-WPREUw | For making CreBC line from enhancer |
| Recombinant DNA reagent | CreStop | This paper | HJP-225-actin5C-Flox-IVS-Syn21myr-tdTomato-p10w | |
| Recombinant DNA reagent | Cre80Tom | This paper | HJP-223-actin5CSyn21Gal80IVS-Syn21myr-tdTomato-p10w | |
| Recombinant DNA reagent | Cre80Tom-GFP | This paper | HJP-224-actin5CSyn21Gal80IVS-Syn21myr-tdTomato-10UAS-mCD8GFP-p10w | |

*Continued on next page*

*Continued*

| Reagent type (species) or resource | Designation | Source or reference | Identifiers | Additional information |
|---|---|---|---|---|
| Recombinant DNA reagent | UAS-Cre$_C$ | This paper | HJP-226-10XUAS-NrdJ-1$^C$-Cre$_C$ | |
| Recombinant DNA reagent | DNE-FRterminator | This paper | HJP-473-DNE > -T2A-Cre$_{AB}$-NrdJ-1$^N$ | |
| Recombinant DNA reagent | Cre80-Kir2.1 | This paper | HJP-472-actin5C-FloxSyn21Gal80-10XUAS-EGFP-Kir2.1 | Genetic reagent (*D. melanogaster*) |
| Recombinant DNA reagent | Cre80-dTrpA1 | This paper | HJP-382-actin5C-FloxSyn21Gal80-10XUAS-dTrpA1 | Genetic reagent (*D. melanogaster*) |
| Chemical compound, drug | Gateway LR Clonase II Enzyme mix | Thermofisher Scientific | 11791100 | |
| Chemical compound, drug | In-Fusion HD Cloning Plus | Takara Bio USA, Inc | 638911 | |
| Chemical compound, drug | pCR8/GW/TOPO TA Cloning Kit with One Shot TOP10 Chemically Competent *E. coli* | Invitrogen Life Technologies | K2500-20 | |
| Chemical compound, drug | pENTR/D-TOPO Cloning Kit, with One Shot TOP10 Chemically Competent *E. coli* | Invitrogen Life Technologies | K240020 | |
| Chemical compound, drug | Q5 High-Fidelity 2X Master Mix | New England Biolabs | M0492S | |
| Chemical compound, drug | Quick Ligation Kit | New England Biolabs | M2200L | |

## *Drosophila* stocks

Vinegar flies of the species *Drosophila melanogaster* were used in this study. Unless otherwise noted, all flies were grown on BDSC Cornmeal Food and maintained at 25°C in a constant 12 hr light–dark cycle. Both male and female progeny of the genotypes indicated in *Supplementary file 2* were used in this study. Previously described fly stocks and their sources are listed in the Key Resources Table. Fly lines generated for this study were made using the DNA constructs described below. Injection of these constructs to produce transgenic flies was carried out by Rainbow Transgenic Flies, Inc (Camarillo, CA). All transgene insertions except the insertion of the DNE-Gal4 were mediated by ΦC31 integrase and placed in the defined attP landing sites indicated in Key Resources Table. Flies made with the DNE-Gal4 were generated by p-element mediated transgenesis. Transgenic flies of the NBE-Cre$_B$ library have transgene insertions on the 2$^{nd}$ chromosome at attP40, while all flies in the NBE-Cre$_C$ library have insertions on the 3$^{rd}$ chromosome at either VK00033 or VK00027.

## Molecular biology

All oligonucleotide and gBlock synthesis was carried out by Integrated DNA Technologies, Inc (Coralville, Iowa), and all final constructs were verified by sequencing (Eurofins Scientific, Louisville, KY or Macrogen Corp, Rockville MD). For routine molecular biology, the following reagents were used according to the manufacturers' supplied protocols: PCR amplification: Q5 High-Fidelity 2X Master Mix #M0492S (New England Biolabs, Ipswich, MA); DNA ligation: Quick Ligation Kit #M2200L (New England Biolabs, Ipswich, MA); Cloning: Gateway LR Clonase II Enzyme mix #11791100 (Thermofisher Scientific, Waltham, MA), and In-Fusion HD Cloning Plus #638911(Takara Bio USA, Inc, Mountain View, CA). gBlocks were used to generate most of the final and intermediate constructs described below, including the DNA fragments encoding the NrdJ-1 and gp41-1 split inteins and the Cre fragments described in the manuscript. DNA sequences of the split inteins were back-translated from the published protein sequences (*Carvajal-Vallejos et al., 2012*) and all sequences were codon biased for *Drosophila*. Sequences of all gBlock fragments and PCR primers are listed in

*Supplementary file 3*. The following reagents, which were used to make several constructs as indicated below, are all described in *Pfeiffer et al. (2010)*: pBPGal80Uw-5, pBPLexA::P65, 10XUAS-IVS-myr::tdTomato, 10XUAS-mCD8::GFP, and pBPGAL80Uw-6.

## Cre80Tom constructs

The indicated Cre80Tom constructs were made stepwise using the described procedures.

### Cre80Tom

Step 1 - Made the intermediate construct 'M1:' an NgoMIV-gBlock013-AatII fragment, an AatII-Gal80-SV40-MfeI fragment (from pBPGal80Uw-5), and an MfeI-gBlock014-KpnI fragment were placed between the NgoMIV and KpnI restriction sites of pBPLexA::P65. Step 2: PCR amplified a KpnI-IVS-StuI-AgeI-myr-tdTomato fragment (Primer71 + Primer72, 10XUAS-IVS-myr::tdTomato as template) and a p10-XbaI fragment (Primer71a + Primer72a, using as template CCAP-IVS-Syn21-KZip$^+$-p10; *Dolan et al., 2017*) and used these to replace the KpnI-XbaI fragment of M1 using In-Fusion HD cloning to make the intermediate construct 'M2.' Gateway cloning of M2 was then performed to add the Actin5C promoter (*Harris et al., 2015*) and get the final Cre80Tom construct.

### Cre80Tom-GFP

Step 1: A 10XUAS-mCD8GFP PCR fragment (template 10XUAS-mCD8::GFP, Primer59+Primer58) was inserted into the unique NdeI site between the mini-white gene and the attB sequence of the M1 vector. Step 2: A KpnI-IVS-Syn21-myr-tdTomato-StuI PCR fragment (Primer75, Primer76, template:10XUAS-IVS-myr::tdTomato) and a StuI-p10-SpeI PCR fragment (primer HJ077, HJ078) were placed between the KpnI and SpeI restriction sites to replace the LexA::P65 fragment and to produce the intermediate construct 'M3' using the In-Fusion HD cloning kit. Step 3: Used Gateway Cloning to add the Actin 5C promoter to produce the *Cre80Tom-GFP*.

### Cre80-dTrpA1 and Cre80-Kir2.1

The sequence between the KpnI and NsiI of M3 (including the IVS-Syn21-myr-tdTomato-p10- and a small part of the mini-white gene) were replaced with gBlock25 by HD-infusion cloning to make the intermediate construct 'M4.' This step removed the tdTomato gene. The BglII-mCD8GFP-XbaI fragment of M4 was replaced with BglII-dTrpA1-XbaI (template: UAS-dTrpA1, gift from Paul Garrity) and BglII-EFGP-Kir2.1-XbaI (template UAS-EGFP-Kir2.1, gift of Sean Sweeney) PCR fragments and then the actin5C promoter was inserted by Gateway cloning to get *Cre80-dTrpA1* and *Cre80-Kir2.1*.

## Split Cre constructs

All split Cre constructs were made by Gateway cloning (LR reaction). Two sets of destination vectors with split Cre components were made: one for use with entry clones containing promoters, and another for entry clones containing enhancers. The 134 NBE entry clones were combined with the latter to make the expression clones used to generate the $Cre_B$ and $Cre_C$ libraries.

To make the $Cre_A$ (HJP-176) destination vectors for use with promoter entry clones, a KpnI-IVS-NheI fragment made from annealed oligonucleotides, a NheI-gBlock012-AgeI gBlocks fragment and an AgeI-PmeI-WPRE-HindIII PCR fragment (amplified from pBPGAL80Uw-6 using PrimerS472 and PrimerS473) were placed between the NheI and HindIII restriction sites of the pBPGw vector (Addgene Plasmid #17574 *Pfeiffer et al., 2008*). Other split Cre destination vectors (i.e. HJP177 ~HJP180; see the Key Resources Table) were made by replacing the NheI-$Cre_A$-gp41-1$^N$-AgeI fragment in $Cre_A$ (HJP-176) with fragments consisting of: NheI-gBlock010-SphI + SphI-gBlock011-AgeI (HJP177), NheI-gBlock008-BsaI+BsaI-gBlock009-AgeI (HJP178), NheI-gBlock007-AgeI (HJP179), or NheI-gBlock010-SphI+SphI-gBlock015-AgeI (HJP180). To create a set of destination vectors for use with enhancer entry clones ('the U-series'), an FseI-DSCP-KpnI synthetic core promoter (*Pfeiffer et al., 2008*) was made from annealed oligos and inserted between the FseI and KpnI restriction sites of each of the destination vectors made for use with promoter entry clones. This produced constructs HJP194 ~196, HJP-207 and HJP-208 (See Key Resources Table).

Prior to the production of transgenic fly lines, the functionality of all Cre constructs was validated in cultured S2 cells by placing the constructs under the control of the Actin5C promoter and testing in appropriate combinations for expression and activity using a floxed reporter construct.

## DNE and NBE entry clones

### DNE

A 2 kb region upstream of the *deadpan* gene previously shown to harbor a NB enhancer by *Emery and Bier (1995)* was Evoprinted (*Yavatkar et al., 2008*) using the sequences of five *Drosophila* species (*D. sechellia*, *D. simulans*, *D. erecta*, *D. yakuba*, and *D. ananasseae*) in addition to *D. melanogaster*. A 607 bp region starting 899 nucleotides 5' of the transcription start exhibited highly conserved sequence blocks containing transcription factor binding sites, including three CAGCTG E-boxes commonly found in other NB enhancers (*Brody et al., 2012*). A PCR fragment containing this 607 bp region was cloned into the Bullfinch Gal4 reporter vector (*Brody et al., 2012*), and the DNE enhancer was made by inserting next to it a previously described mutant *nerfin-1* enhancer with two adjacent bp substitutions (G→C and T→C) that were shown to expand the pattern of NB expression (*Kuzin et al., 2011*). The mutant nerfin-1 enhancer was amplified by PCR from pCRII-TOPO (Thermofisher Scientific, Waltham, MA) and is separated from the *dpn* enhancer by 10 bp of DNA sequence from the pCRII-TOPO vector, including the EcoRI site that was used to insert this enhancer adjacent to the *dpn* enhancer. A DNE vector for use in Gateway cloning was made by transferring the DNE enhancer into the pENTR-D-TOPO entry clone as a PCR fragment (primers: DNE-Sense and DNE-Antisense) using the pENTR/D-TOPO Cloning Kit.

Most of the neuroblast-active enhancers used to make the NBE entry clones were from the JFRC Flylight Collection (*Pfeiffer et al., 2008*).Candidate Flylight enhancers were selected either on their previous identification as embryonic neuroblast enhancers (active in subset of neuroblasts) (*Manning et al., 2012*) or on the presence of expression in NBs in the 3rd instar CNS as determined by visual inspection of the expression patterns at the Flylight website (https://www.janelia.org/project-team/flylight). To verify NB expression of the latter NBEs, we applied the same method used to examine Cre activity in the single ventral nerve cord lineage labeled by the R59E09-Gal4 line (*Figure 1—figure supplement 2D*). Flylight Gal4 lines made with the candidate enhancers were pre-screened by crossing them to flies containing the CreStop (HJP225) and UAS-Cre$_C$ (HJP266) constructs described below with the following genotype: w, DNE-Cre$_B$(attP8); UAS-Cre$_C$(attP40); CreStop (i.e. actinˆSTOPˆtomato(attP2)), DNE-Cre$_A$(VK00027). CNS preparations of the progeny (third instar larvae or adults) were examined for tdTomato expression in NB clones. Selected JFRC Neuroblast active enhancers (NBEs) with 'sparse' expression in neuroblasts were amplified by PCR or synthesized when PCR failed (Epoch Life Science, Inc, Missouri City, TX) and cloned into either the pCR8-GW-TOPO or pENTR-D-TOPO donor vectors. Primers listed at the Flylight website were used to amplify most JFRC NBEs using genomic DNA from either y; cn bw sp [gift from James A. Kennison] or Canton S wildtype flies as template.

The cas-8 and CG7229-5 enhancers (*Brody et al., 2012*; *Kuzin et al., 2012*) were synthesized as gBlock fragments and cloned by HD-Infusion cloning. The pdm-2-37a (*Ross et al., 2015*), cas-5 (*Kuzin et al., 2012*), danR-1, svp-29, and tll-15 enhancer sequences (gifts from Jermaine Ross) were amplified as PCR fragments from plasmids and placed between the NotI and AscI sites of pENTR/D-TOPO vector. The entry clones for the *stg-14* (*Wang et al., 2014*) and *otd* (*Asahina et al., 2014*; *Gao and Finkelstein, 1998*) enhancers have been previously described.

### FRTerminator

This construct (HJP-473) was made as follows: an AvrII and PmeI flanked DNA fragment (including partial nerfin-1 enhancer, FRT and Syn21-flipase-T2A-Cre$_A$-gp41-1$^N$-AgeI-FRT) were synthesized (Epoch Life Science, Inc, Missouri City, TX) and put between the AvrII and PmeI restriction sites of DNE-CreA-gp41-1$^N$. The resulting construct can be used in place of DNE-Cre$_A$ in Step 2 SpaRCLIn screens. It was tested, but its use is not described in this manuscript. This construct was used as an intermediary to make the final FRTerminator construct by inserting gBlock-043 (part of the CreAB sequence and Nrdj-1N) into its SbfI and AgeI restriction sites using the In-Fusion HD cloning technique.

### Other constructs

Two constructs were used to pre-screen candidate enhancers driving Gal4 expression. These included CreStop (HJP225) and UAS-Cre$_C$ (HJP266). The CreStop construct was made using a Ngo-MIV-loxP-hsp70 terminator-MluI gBlock to replace the loxP-Gal80 in Cre80Tom (HJP223) by In-

Fusion HD cloning. UAS-Cre$_C$ was made by cloning a NotI-NrdJ-1$^C$-Cre$_C$-XbaI PCR fragment (Primer116 and Primer117; Cre$_C$ as template) between the NotI and XbaI sites of pJFRC1-10XUAS-mCD8::GFP using the In-Fusion HD cloning technique.

## Immunostaining and image acquisition

Embryos were prepared and immunostained as described by *Lécuyer et al. (2008)*. Excised nervous system whole mounts were prepared from wandering third-instar larvae or adults after dissection into PBS and fixation in 4% paraformaldehyde in PBS for 20–30 min. Immunostaining was done with the antibodies listed in the Key Resources Table at the indicated dilutions. For confocal imaging, all tissues were attached to poly-L-lysine coated cover glass and mounted in Vectashield (Vector Laboratories, Burlingame, CA) prior to imaging with a Nikon C-2 confocal microscope. Z-series were acquired in 0.85 µm increments using a 20X objective using 488 nm, 543 nm or 633 nm laser emission lines for fluorophore excitation. The images shown are maximal projections of volume rendered z-stacks of confocal sections taken through the entire nervous system. NB expression of Gal4 driven by the DNE enhancer was examined in embryonic fillets by in situ hybridizations as previously described (*Ross et al., 2015*).

## Proboscis extension assay

Flies assayed for proboscis extension were raised at 25°C until the white prepupa stage and then transferred to 18°C until the time of testing. For neuronal activation using dTrpA1, the chambers were placed on the surface of the Echotherm Chilling/Heating Dry Bath IC25 (Torrey Pines Scientific, Inc, Carlsbad, CA) at 31°C. For the Step 1 SpaRCLIn screen, approximately a dozen adult flies (3–10 d old) of each genotype were placed in glass TriKinetics tubes (3 mm inner diameter; TriKinetics Inc, Waltham, MA) and videorecorded at 31°C for 3 min using a Sony NEX-VG10 videocamera. Proboscis extension behavior was analyzed from these recordings. If two or more flies exhibited robust, full-length extension, the cross was scored as positive. For the Step two tripartite SpaRCLIn screen, two flies at a time (one male and one female) were videorecorded together in glass minichambers (0.3 cm diameter X 0.7 cm length) for 3 min at 18°C followed by 3 min at 31°C. Flies were subjected to these temperature transitions twice and proboscis extension behavior was analyzed following the recording. The criteria for positive proboscis extension was three or more bouts of full proboscis extension in both tests. For the FRTerminator behavior experiments flies were subjected to only one test. Flies used to make the videos included in the manuscript were back-mounted on a 200 uL pipette tip with 5-Minute-Rapid-Curing, General Purpose Adhesive Epoxy (ITW polymers Adhesive, Danvers, MA) and placed just above the heating plate, which was adjusted to apply temperature changes.

## Acknowledgements

We thank Matthew Roberts, Andrew Laczarchik, and Ana Cardenas for technical help in generating the NBE-Split Cre lines described in this study. We also thank Sean Sweeney, David Anderson, and Jermaine Ross for plasmid DNA; Paul Garrity and James Kennison for fly lines; and Kiichi Watanabe for sequence details regarding the *otd* Gateway entry clone. This work was supported by the Intramural Research Programs of the National Institute of Mental Health (ZIA-MH002800, BHW) and the National Institute of Neurological Disease and Stroke (ZIA-NS002820-26). We further thank the Bloomington *Drosophila* Stock Center (NIH P40OD018537) for many of the fly stocks used in this study, and members of the White lab for insightful comments on the manuscript.

## Additional information

### Funding

| Funder | Grant reference number | Author |
| --- | --- | --- |
| National Institute of Mental Health | ZIA-MH002800 | Benjamin H White |
| National Institute of Neurological Disorders and Stroke | ZIA-NS002820-26 | Ward F Odenwald |

The funders had no role in study design, data collection and interpretation, or the decision to submit the work for publication.

## Author contributions
Haojiang Luan, Conceptualization, Resources, Formal analysis, Supervision, Funding acquisition, Validation, Investigation, Visualization, Methodology; Alexander Kuzin, Conceptualization, Supervision, Validation, Investigation, Visualization, Methodology; Ward F Odenwald, Resources, Funding acquisition, Investigation, Visualization; Benjamin H White, Conceptualization, Resources, Formal analysis, Supervision, Funding acquisition, Investigation, Visualization, Methodology

## Author ORCIDs
Benjamin H White (iD) https://orcid.org/0000-0003-0612-8075

## Decision letter and Author response
Decision letter https://doi.org/10.7554/eLife.53041.sa1
Author response https://doi.org/10.7554/eLife.53041.sa2

# Additional files

## Supplementary files
• Supplementary file 1. List of Neuroblast Enhancers (NBEs). The table lists all NBEs used in this study, their sources, and the names of the corresponding CreB and CreC constructs made with them.

• Supplementary file 2. Fly genotypes by Figure. The table indicates the genetic crosses performed to generate the flies used in each figure of the paper (parental genotypes are shown).

• Supplementary file 3. Primer and gBlock sequences. The table lists the primer and gBlock sequences used to generate constructs described in this paper.

• Supplementary file 4. Plasmid DNA sequences of all constructs made for this paper.

• Transparent reporting form

## Data availability
Data generated during this study are included in the manuscript and supporting files. Source data files have been provided for Figure 2I.

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
