## [Decision Letter]

Thank you for submitting your article "Cre-assisted fine-mapping of neural circuits using orthogonal split inteins" for consideration by *eLife*. Your article has been reviewed by two peer reviewers, and the evaluation has been overseen by K VijayRaghavan as the Senior and Reviewing Editor. The following individual involved in review of your submission has agreed to reveal their identity: Chris Q Doe (Reviewer #1).

The reviewers have discussed the reviews with one another and the Reviewing Editor has drafted this decision to help you prepare a revised submission.

Summary:

Haojiang Luan, Ben White, and their colleagues gifted *Drosophila* neurobiologists with the bipartite split-GAL4 system, which has become fundamental in targeting genetic manipulations to sparse neuronal subsets. So, it is with great anticipation that one approaches their creative new method for further refining GAL4 expression patterns. Of course, several methods of refinement already exist, such as intersecting split-GAL4 patterns with FLP, GAL80, or killer-Zip constructs. The novelty in the method they report here is in using neuroblast-specific enhancers as the third intersecting component, rather than additional enhancers expressed in differentiated neurons. This idea is not new but has proven difficult to implement because of the lack of well-defined neuroblast enhancers. Luan and colleagues partially solve this problem by making neuroblast enhancers out of bi- and tri-partite intersections of multiple enhancers, one of which is a synthetic pan-neuroblast enhancer. To do this, they exploit the split intein system to reconstruct a functional Cre recombinase, which in turn excises a ubiquitous GAL80 cassette to allow GAL4 expression within the progeny of the targeted lineage. The method is perhaps somewhat cumbersome, given the large number of transgenes involved, but this should be manageable, particularly if one could generate a panel of stocks, each of which containing all of the split-Cre components needed to target a specific lineage.

The practical utility of this method will depend principally on (a) the coverage and specificity of the collection of neuroblast Cre lines, and (b) the reproducibility of the resulting expression patterns. With respect to coverage and specificity, the collection of lines here is a good starting point, though as the authors acknowledge, it needs some improvement. They envision, not unreasonably, that this will become a community effort over time, as users identify new neuroblast enhancers and gradually improve the collection. The question is whether, as the reagents currently stand, a sufficient utility threshold is reached so that the community will indeed enthusiastically take up this approach and gradually improve the reagents. The first split-GAL4 system was not perfect and few useful lines were initially generated. But it clearly filled a void and was effective enough that it was rapidly embraced and continuously improved. It is hard to see the split-Cre system catching on in a similar way. Partly this is because of the success of split-GAL4, and the existence of alternative methods for further refinement. The need is simply not as great as it was for split-GAL4. But it is also because of the problem of poor reproducibility of the split-Cre system.

Because of the variable expression patterns obtained with the split-Cre system, circuit mapping using this system functions similarly to the existing stochastic methods for restricting expression, i.e. large numbers of individuals need to be tested behaviorally and then analysed histologically to determine which cells were targeted, fly-by-fly. This is how the authors use it in the examples they provide here. One would need to consider on a case-by-case basis which is the more suitable approach. Split-Cre likely involves more crosses but fewer flies in order to get a convincing correlation, but may be more prone to failure due to coverage issues or the inability to discriminate between cells in the same lineage. A simple hs-FLP approach would be easier genetically, requiring fewer stocks and crosses, but perhaps require examining more individuals. The best choice will likely depend on the complexity of the initial pattern, but often standard hs-FLP approach is likely to still be the easiest and most reliable.

The strengths of the work are also notable. One is the clearly demonstrated ability to narrow a Gal4 pattern by intersecting it with two or three Cre patterns. Another is using the sparse patterns to identify neurons that can induce a specific behavior; leveraging the stochasticity of the expression to correlate expression with behavioral phenotype is really turning lemons (variability) into lemonade (stronger neuron-behavior correlations). Yet another strength is the creation of new transgenics to simplify use of the system (for the authors and all who follow). Very thoughtful.

Overall, there are some very important concerns that need to be addressed before acceptance.

Essential revisions:

1) For the above reasons, our enthusiasm for the split-Cre approach is somewhat muted. It is clever and potentially very useful, but we think it still needs further development before it will be enthusiastically taken up by the community. We suggest, reluctantly, that the authors delve further into the reasons for the variability. It might be easier to do this using a simple Cre-dependent reporter (which they have), rather than indirectly by dissecting some GAL4 expression pattern (as they do with TH-GAL4 here). A more robust method, even with only a limited initial set of neuroblast-specific Cre lines, would certainly be a useful tool for the community.

2) It is not clear that the NBEs are really neuroblast-specific. Data supporting this conclusion could be readily obtained by staining the NBE-Gal4 UAS-GFP lines with GFP (for the gal4 pattern) and Dpn or Wor (for neuroblasts); there should be only one double positive cell in each hemisegment of the VNC or brain lobe.

3) It is unclear how reproducible the NBE pattern is when a NBE-Cre(AB) is inserted at different landing sites other than attP2. Are there data on this point?

4) How early does the Cre reconstitution start to work? Staining for temporal factor expression may help to reveal the onset time.

---

## [Author Response]

Summary:Haojiang Luan, Ben White, and their colleagues gifted Drosophila neurobiologists with the bipartite split-GAL4 system, which has become fundamental in targeting genetic manipulations to sparse neuronal subsets. So, it is with great anticipation that one approaches their creative new method for further refining GAL4 expression patterns. Of course, several methods of refinement already exist, such as intersecting split-GAL4 patterns with FLP, GAL80, or killer-Zip constructs. The novelty in the method they report here is in using neuroblast-specific enhancers as the third intersecting component, rather than additional enhancers expressed in differentiated neurons. This idea is not new but has proven difficult to implement because of the lack of well-defined neuroblast enhancers. Luan and colleagues partially solve this problem by making neuroblast enhancers out of bi- and tri-partite intersections of multiple enhancers, one of which is a synthetic pan-neuroblast enhancer. To do this, they exploit the split intein system to reconstruct a functional Cre recombinase, which in turn excises a ubiquitous GAL80 cassette to allow GAL4 expression within the progeny of the targeted lineage. The method is perhaps somewhat cumbersome, given the large number of transgenes involved, but this should be manageable, particularly if one could generate a panel of stocks, each of which containing all of the split-Cre components needed to target a specific lineage.The practical utility of this method will depend principally on (a) the coverage and specificity of the collection of neuroblast Cre lines, and (b) the reproducibility of the resulting expression patterns. With respect to coverage and specificity, the collection of lines here is a good starting point, though as the authors acknowledge, it needs some improvement. They envision, not unreasonably, that this will become a community effort over time, as users identify new neuroblast enhancers and gradually improve the collection. The question is whether, as the reagents currently stand, a sufficient utility threshold is reached so that the community will indeed enthusiastically take up this approach and gradually improve the reagents. The first split-GAL4 system was not perfect and few useful lines were initially generated. But it clearly filled a void and was effective enough that it was rapidly embraced and continuously improved. It is hard to see the split-Cre system catching on in a similar way. Partly this is because of the success of split-GAL4, and the existence of alternative methods for further refinement. The need is simply not as great as it was for split-GAL4. But it is also because of the problem of poor reproducibility of the split-Cre system.Because of the variable expression patterns obtained with the split-Cre system, circuit mapping using this system functions similarly to the existing stochastic methods for restricting expression, i.e. large numbers of individuals need to be tested behaviorally and then analysed histologically to determine which cells were targeted, fly-by-fly. This is how the authors use it in the examples they provide here. One would need to consider on a case-by-case basis which is the more suitable approach. Split-Cre likely involves more crosses but fewer flies in order to get a convincing correlation, but may be more prone to failure due to coverage issues or the inability to discriminate between cells in the same lineage. A simple hs-FLP approach would be easier genetically, requiring fewer stocks and crosses, but perhaps require examining more individuals. The best choice will likely depend on the complexity of the initial pattern, but often standard hs-FLP approach is likely to still be the easiest and most reliable.The strengths of the work are also notable. One is the clearly demonstrated ability to narrow a Gal4 pattern by intersecting it with two or three Cre patterns. Another is using the sparse patterns to identify neurons that can induce a specific behavior; leveraging the stochasticity of the expression to correlate expression with behavioral phenotype is really turning lemons (variability) into lemonade (stronger neuron-behavior correlations). Yet another strength is the creation of new transgenics to simplify use of the system (for the authors and all who follow). Very thoughtful.Overall, there are some very important concerns that need to be addressed before acceptance.

We thank the reviewers for your thoughtful comments on our submitted manuscript. You raised concerns that we had not previously appreciated and alerted us to unintended omissions. To meet the reviewers’ requests for essential changes, we have made numerous changes, which we believe strengthen the manuscript considerably. These changes and our point-by-point responses are detailed below, but we wished to start by addressing the two overarching reservations raised about the SpaRCLIn technique. The first of these relates to its efficacy, and second to its ease of use relative to existing methods, particularly Split Gal4.

With regard to efficacy, major concerns were raised about “(a) the coverage and specificity of the collection of neuroblast Cre lines, and (b) the reproducibility of the resulting expression patterns.” Prompted by these concerns we have introduced new data regarding the spatiotemporal expression of our NB enhancers and its variability. With regard to the neuroblast-specific DNE: we show that DNE-driven Cre activity is evident as early as embryonic stage 8 and is widespread in the developing CNS by stage 13 (subsection “Development of Bipartite and Tripartite Split Cre Recombinases”, fifth paragraph; Figure 1—figure supplement 2C in the revised manuscript). As requested by the reviewers, we also demonstrate that the DNE drives Cre fragment expression early enough to label progeny of a defined NB lineage identified by Lacin and Truman, 2016 (Figure 1—figure supplement 2D in the revised manuscript). Although we cannot rule out the absence of Cre activity in the earliest born neurons, these new data suggest that Gal80 excision occurs relatively early. We anticipate that coverage of adult neurons, most of which are born during larval neurogenesis is much better, except for those in the optic lobe where the DNE is not active as previously noted.

The reviewers point out that the best possible coverage is likely afforded by standard hs-Flp methods and that such methods are easier to use. The downside of these methods, however, is their very limited reproducibility. The reproducibility achieved with almost all NBE lines of SpaRCLIn is at least an order of magnitude better than that of Flp-out Gal80 methods (approximately 3% for expression in a neuron of interest in a study in which we participated, Flood et al., 2013). As our dissection of both the TH- and Rk-Gal4 lines shows, the NBE lines with the best reproducibility will express in a given neuron in upwards of 80% of progeny. Although this falls short of complete reproducibility, it does allow an end-user to identify a neuron of interest in a screen and repeatedly return to it for further characterization with some confidence. In contrast, with hs-Flp methods, one is forced to search the entire haystack (i.e. Gal4 expression pattern) again for the desired needle (i.e. neuron) for every new manipulation (i.e. UAS effector) one uses. The utility of hs-Flp methods thus steeply declines if one wants to do more than simply identify a neuron based on one criterion. As the reviewers point out, end users will want to decide which method to choose depending on their needs, but we believe that SpaRCLIn offers sufficient advantages to be an attractive option in several contexts.

Under what circumstances might an end-user choose to use SpaRCLIn over Split Gal4? We recommend using it when Split Gal4 can’t (or seems unlikely to) deliver the desired refinement in expression. We agree with the reviewers that Split Gal4 and other existing tools have revolutionized *Drosophila* neurobiology and we appreciate their kind comments regarding our contributions to this effort. However, in many cases even the best available tools don’t restrict expression sufficiently to permit assignment of functional identities to particular neurons. This is reflected in the question we most frequently get from potential end-users regarding SpaRCLIn: Does it work with Split Gal4? The coverage of Split Gal4—using the thousands of lines generously generated, characterized, and made available by the Rubin and Dickson labs—is, like that of SpaRCLIn, incomplete (estimated at 75% by Dionne et al., 2018), and finding useful combinations – while facilitated by fine registration tools developed by several labs – still generates “no more that 5%” of intersections without ectopic expression (Dionne et al., 2018). Under these circumstances, end-users will benefit from additional strategies and SpaRCLIn – with its distinct differences from other commonly used strategies – should prove useful.

Essential revisions:1) For the above reasons, our enthusiasm for the split-Cre approach is somewhat muted. It is clever and potentially very useful, but we think it still needs further development before it will be enthusiastically taken up by the community. We suggest, reluctantly, that the authors delve further into the reasons for the variability. It might be easier to do this using a simple Cre-dependent reporter (which they have), rather than indirectly by dissecting some GAL4 expression pattern (as they do with TH-GAL4 here). A more robust method, even with only a limited initial set of neuroblast-specific Cre lines, would certainly be a useful tool for the community.

We share the reviewers’ interest in the cause(s) of variability and have added new information (subsection “Development of Bipartite and Tripartite Split Cre Recombinases”, last paragraph) and a new figure (Figure 1—figure supplement 3) to the revised manuscript regarding this point. As noted in the Discussion of the original manuscript, erratic expression may derive in part from intrinsic stochasticity of the individual NB enhancers. As the reviewers pointed out, this possibility can be investigated using our Cre-dependent Act5C^Gal80^tdTomato (i.e. CreStop) reporter. We had previously confirmed NBE-driven expression in larval CNS preparations using this reporter (see Figure 1—figure supplement 1), and we have now examined multiple preparations (n= 4 on average) for each of the 134 NBEs to assess their variability in DNE-Cre_AB_∩NBE-Cre_C_ > CreStop crosses (in addition to several similar DNE-Cre_A_∩DNE-Cre_C_∩NBE-Cre_B_ crosses, see Essential revision 3 below). In general, we find that patterns for any given NBE are similar across preparations, with examples presented in a new figure (i.e. Figure 1—figure supplement 3A-D). Although some variation is almost certainly present, the large size of the patterns and our inability to reliably identify identical neurons and lineages across preparations prevented a more detailed analysis. As noted in the revised Discussion (subsection “Variability of SpaRCLIN expression”, first paragraph), our results suggest that large-scale variability in the activity of individual NBEs is unlikely to explain all of SpaRCLIn’s stochasticity in Step 2 crosses and that further analysis will be required to determine its source. We also now provide a somewhat more extended discussion of the possible sources with addition of the following material: “If a Cre component is only weakly expressed, or expressed late during neurogenesis, a limited amount of active, full-length Cre may be produced and excision of Gal80 may be sporadic in the expressing neuroblasts. […] Finally, because the efficacy of Cre-mediated excision depends on the distance between the loxP sites flanking the excised fragment, it may prove possible to increase the efficacy of excision – and thus reduce stochasticity of expression – using strategies that decrease the distance between the loxP sites flanking the excised fragment.”

2) It is not clear that the NBEs are really neuroblast-specific. Data supporting this conclusion could be readily obtained by staining the NBE-Gal4 UAS-GFP lines with GFP (for the gal4 pattern) and Dpn or Wor (for neuroblasts); there should be only one double positive cell in each hemisegment of the VNC or brain lobe.

We realized in reading this comment that we had provided too little information about the sources and properties of our NBEs. Few (perhaps none) of them are “neuroblast-specific.” This is why the Split Cre approach was required. Expression of one Cre component had to be driven by a bona fide NB-specific enhancer (viz. the “DNE”) to delimit Cre activity to NBs. Importantly, however, all of our NBEs are active in neuroblasts (in addition to other cell types). This may have been unclear because we described the procedure for selecting and testing previously unpublished NBEs only in the Materials and methods and not in the body of the paper. While the majority of our NBEs were selected from previously published sets (88 of the 134 NBEs are from Manning et al., 2012), approximately one-quarter (38) were identified by a method similar to that suggested by the reviewers. We used NBE-Gal4 lines from the Janelia FlyLight collection with probable expression in larval NBs to drive a UAS-Cre_C_ construct together with Cre_A_ and Cre_B_ fragments driven by DNE. Expression in NB clones was evaluated using the CreStop reporter. The main text of the revised manuscript now clearly indicates the origin of the NBEs (subsection “Development of Bipartite and Tripartite Split Cre Recombinases”, third paragraph), with all sources cited in a new column in the revised Supplementary file 1.

3) It is unclear how reproducible the NBE pattern is when a NBE-Cre(AB) is inserted at different landing sites other than attP2. Are there data on this point?

We are somewhat confused by this point. We do not describe an NBE-Cre_AB_ construct/line in our paper. The only Cre_AB_ line is the one used for Step 1 screening where expression of the Cre_AB_ fragment is driven by the DNE. The DNE-Cre_AB_ insertion in this case is into the JK22C attP landing site on Chromosome II. This site was chosen because of its previously demonstrated fidelity of expression compared with attP2 (Knapp et al. Genetics. 2015, 199: 919–934, see Figure 2E).

We wonder, however, if the reviewer isn’t referring to the reproducibility of expression produced by NBE-Cre_C_ and NBE-Cre_B_ lines. The former (with insertions on Chromosome III: attP VK00033 or VK00027) are used in Step 1 screens to select suitable NBE-Cre_B_ lines (with insertions on Chromosome II: attP40) for use in Step 2 screening. Ensuring the similarity of expression between NBE-Cre_B_ and NBE-Cre_C_ lines made with identical NBEs is thus important. This is an issue that we had investigated, but did not explicitly address in the original manuscript. The revised manuscript, however, incorporates some of our previous findings together with more recent results (subsection “Development of Bipartite and Tripartite Split Cre Recombinases”, last paragraph).

The larval CNS expression patterns produced by all NBE-Cre_B_ lines had been visually inspected by fluorescence microscopy and compared to the corresponding patterns obtained for NBE-Cre_C_ lines (characterized as described in Essential Revision 1). The patterns for a given enhancer inserted at the two different attP landing sites were generally similar, although differences in expression levels between the two types of lines made detailed comparison difficult. (This is because the NBE-Cre_C_ lines were tested using the bipartite system, while NBE-Cre_B_ lines were necessarily tested using the tripartite system, which yields lower expression.) Nevertheless, we flagged 24 NBE-Cre_B_ lines whose expression pattern differed overtly in one or more respects from the corresponding NBE-Cre_C_ lines. We have now tested several of these lines more systematically and provide an example of one such NBE in the revised manuscript in Figure 1—figure supplement 3C, D where it is compared with an example of an NBE that produced similar patterns (Figure 1—figure supplement 3A, B). The revised manuscript also now alerts readers to the 24 potentially problematic lines, which are marked with asterisks in Supplementary file 1. A footnote in the table of Supplementary file 1 states that users who identify one of these NBE-Cre_C_ lines as a “positive hit” in a Step 1 screen may want to continue using this line in the subsequent Step 2 screen (if possible) to preserve its expression pattern.

4) How early does the Cre reconstitution start to work? Staining for temporal factor expression may help to reveal the onset time.

As noted above, we now present data in the revised manuscript (Figure 1—figure supplement 2C) that suggests a relatively early onset of Cre reconstitution. Since we assayed Cre-mediated tdTomato expression and because this expression depends on multiple time-dependent steps (i.e. expression of the split Cre fragments, their reconstitution into functional recombinase, excision of Gal80 by the recombinase, and expression of sufficient tdTomato reporter to see a signal) our data set an upper bound on the time of Cre reconstitution.